# Aperiodic spin chains at the boundary of hyperbolic tilings

**Pablo Basteiro, Rathindra Nath Das, Giuseppe Di Giulio⋆ and Johanna Erdmenger**

Institute for Theoretical Physics and Astrophysics and
Würzburg-Dresden Cluster of Excellence ct.qmat,
Julius-Maximilians-Universität Würzburg,
Am Hubland, 97074 Würzburg, Germany

⋆ giuseppe.giulio@physik.uni-wuerzburg.de

## Abstract

In view of making progress towards establishing a holographic duality for theories defined on a discrete tiling of the hyperbolic plane, we consider a recently proposed boundary spin chain Hamiltonian with aperiodic couplings that are chosen such as to reflect the inflation rule, i.e. the construction principle, of the bulk tiling. As a remnant of conformal symmetry, the spin degrees of freedom are arranged in multiplets of the dihedral group under which the bulk lattice is invariant. For the boundary Hamiltonian, we use strong-disorder RG techniques and evaluate correlation functions, the entanglement entropy and mutual information for the case that the ground state is in an aperiodic singlet phase. We find that two-point functions decay as a power-law with exponent equal to one. Furthermore, we consider the case that the spin variables transform in the fundamental representation of $SO(N)$, leading to a gapless system, and find that the effective central charge obtained from the entanglement entropy scales as $\ln N$, reflecting the number of local degrees of freedom. We also determine the dependence of this central charge on the parameters specifying the bulk tiling. Moreover, we obtain an analytical expression for the mutual information, according to which there is no phase transition at any finite value of the distance between the two intervals involved.

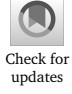

# 1  Introduction

The holographic principle [1, 2], together with its most well-understood realization in the Anti-de Sitter/Conformal Field Theory (AdS/CFT) correspondence [3–5], have fundamentally changed the paradigm of high-energy physics by introducing a dictionary between gravitational theories in $d+1$-dimensions and $d$-dimensional strongly coupled CFTs. Within AdS/CFT, quantum information measures such as entanglement entropy [6–8], mutual information [9] as well as other correlation measures such as Rényi entropies and relative entropy [10–12] have played a ubiquitous role in deciphering and expanding the holographic dictionary. More recently, motivated in part by experimental realizations of hyperbolic space [13–15], the program of *discrete holography* [16–27] has started the development of holographic dualities based on discrete spaces, such as graphs, tilings and lattices. Mostly focusing on the prototypical case suggested by the $AdS_3/CFT_2$ duality, and in particular, by a constant time slice of $AdS_3$, a central ingredient in the majority of these approaches are regular hyperbolic tilings [28–30] that provide a discretization of two-dimensional hyperbolic spaces. Characterized by their so-called Schläfli symbol $\{p, q\}$, denoting a tiling with $q$ regular $p$-gons meeting at each vertex, these tessellations retain in general a large number of symmetries from the continuous case, making them a promising candidate for a discrete geometry. A fascinating feature of $\{p, q\}$ tilings is that they can be constructed systematically through replacement rules in a procedure denoted as *inflation* [31]. Remarkably, the inflation construction induces an aperiodic structure on the boundary of finite hyperbolic tilings.

In [25], this aperiodic structure was exploited to propose an explicit Hamiltonian as a potential boundary theory. The Hamiltonians of these models have been defined in such a way that the nearest-neighbor couplings of the spins follow exactly the aperiodic sequence induced by the inflation rule at the boundary. This leads to an aperiodically modulated sequence of couplings. More specifically, in [25] a spin-1/2 $SU(2)$ aperiodic XXX chain was investigated via strong-disorder renormalization group (SDRG) techniques [32, 33] to obtain its ground state and the correlations present therein. It was found that under certain assumptions, the ground

state of this spin chain is given by an *aperiodic singlet phase* (ASP), in which it factorizes into the product of singlet states between two spins at arbitrarily large and aperiodically distributed distances. Moreover, a large class of $\{p, q\}$ modulations was found to require the application of at most two decimation steps in the SDRG. In turn, this allows to identify two different families of singlets, depending on whether they are obtained by an even or an odd number of decimation SDRG steps. The entanglement entropy of an interval was found to exhibit a piece-wise linear dependence on the sub-system size, together with a logarithmic envelope that matches the predictions from CFT computations [34,35]. The discretization parameters were found to enter only in the multiplicative pre-factor of this envelope, which was identified with an effective central charge, following the spirit of [36]. Moreover, an exact tensor network (TN) realization of the ground state of these aperiodic Hamiltonians was obtained. This TN graph provides a discrete hyperbolic geometry realization of this ground state.

In this work, we advance the study of these aperiodic spin chains with modulation given by the boundary aperiodic sequence obtained from the inflation rule for a $\{p, q\}$ bulk tiling. An interesting question, also in the view of holographic dualities, is whether we can access a regime characterized by a large number of spin degrees of freedom in aperiodic chains. This is motivated by continuous AdS/CFT, where the large $N$ limit ($N$ is associated with the gauge group $SU(N)$ of the boundary theory) is essential for the required saddle-point approximation on the gravity side. In order to achieve a similar feature, we discuss two symmetry-based constructions. The first one is inspired by the fact that in continuous AdS/CFT, the operators of the CFT at the boundary transform covariantly under the irreducible representations (irreps) of the isometry group of the bulk spacetime, which is the conformal group. In our discrete setting [25], after radial truncation, the symmetry of $\{p, q\}$ tilings is the dihedral group $D_n$, with $n = p, q$, which is the symmetry group of a regular polygon.[1] Therefore, we define a spin chain where the local spin degrees of freedom are arranged into multiplets of $D_n$ and transform covariantly under this group. The second approach consists in studying aperiodic spin chains with $SO(N)$ global symmetries. We consider $SO(N)$ instead of $SU(N)$ in order to guarantee a gapless phase [42,43] and the existence of the ASP.[2] Moreover, we consider a global symmetry, instead of the usual gauge symmetry considered in holography [47], in line with the standard analysis of spin chains. We extend the SDRG procedure and the derivation of the ASP to this class of models. To the best of our knowledge, spin chains with these symmetries have never been considered before in presence of aperiodic modulations.

For chains with $SO(N)$ symmetry, we may consider the limit $N \to \infty$. Interestingly, we find that the effective central charge obtained from the entanglement entropy diverges as $\ln N$ when $N \to \infty$, reflecting the property of encoding information about the number of degrees of freedom. Although motivated by the AdS/CFT correspondence, we are aware that the results we discuss in this manuscript are not expected to share any feature observed in continuous holographic setups. Indeed, here we do not consider any notion of strong coupling, while this is a well-known aspect of standard continuous holographic boundary theories such as the SYK model [48–50]. Moreover, the investigated large $N$ limit is essentially different from the large $N$ limit in usual AdS/CFT, given that $SO(N)$ is a global symmetry and the spin variables in the fundamental rather than in the adjoint representation of $SO(N)$. Despite these differences, we expect the results to be useful in view of improving the understanding of the discrete holographic setup developed in [25], and to serve as a basis for further work towards establishing a complete duality that involves also a dynamical bulk theory.

---

[1]For infinite tilings and excluding reflection transformations, the corresponding symmetry groups of $\{p, q\}$ tilings are an example of so-called *Fuchsian groups*. These have been investigated in the context of condensed matter physics and crystallography in [37–41].

[2]For a holographic Kondo model involving an $SU(N)$ spin defect see [44–46].

For the aforementioned spin chains with dihedral symmetry, $D_n$ is a spacetime symmetry of the model, which may be thought of as a remnant of the conformal symmetry of the continuous case. Crucially, however, the discretization introduces a further parameter $n = p, q$ that can be tuned. This parameter is not present in conformal algebra. It allows us to investigate the limit $n \to \infty$, which has no direct counterpart in continuous AdS/CFT. Given that the number of irreps of $D_n$ is proportional to $n$, we find that the effective central charge is rescaled by a factor linear in $n$. This implies that, for a specific class of modulations, we can access a regime of large central effective charge.

The three main results of this paper are summarized as follows. First, for aperiodic coupling modulations associated with hyperbolic $\{p, q\}$ tilings, we generalize the results of [51] on correlation functions to two families of singlets that arise in the SDRG procedure. For spins belonging to the same singlet, we find that their two-point correlation function in aperiodic spin chains with ASPs as their ground state exhibits a power-law decay in the spin separation, with an exponent equal to one. The pre-factors of this correlation function depend on $p$ and $q$ and on the SDRG family of the singlet under consideration. Second, we extend the analysis in [25] on the entanglement entropy of a block of spins in ASPs by deriving exact analytical expressions for the non-universal additive constants of the enveloping functions. Exploiting these results, we compute, for the first time, the mutual information in ASPs. For the case of adjacent intervals, we again find a piece-wise linear behavior with a logarithmic enveloping function, reproducing the functional dependence found in CFT calculations [52]. In the case of disjoint intervals, we find a piece-wise linear behavior but no enveloping functions. Additionally, considered as a function of the distance $d$ between the sub-systems, the mutual information exhibits a decaying behavior, vanishing for finite ranges of $d$ that are separated by peaks of non-vanishing amplitude. In particular, we do not observe a phase transition at any finite value $d_c$ such that the mutual information vanishes for $d > d_c$. This is in contrast to the behavior found in continuous AdS/CFT [9]. Third, we extend the SDRG to $SO(N)$-invariant aperiodic Hamiltonians with spin DOFs in the fundamental representation of $SO(N)$. Computing the entanglement entropy of an interval in this chain, we obtain an effective central charge as a function of $N$. In fact, we find that this effective central charge grows with $\ln N$ when $N \to \infty$.

This paper is organized as follows. In Sec. 2 we discuss the main properties of the regular hyperbolic $\{p, q\}$ tilings. In particular, we focus on the symmetries of these discrete geometries and on the aperiodic structure at their boundaries. In Sec. 3 we review the strong-disorder renormalization group (SDRG) in aperiodic spin chains and the aperiodic singlet phase (ASP) together with its entanglement properties. For aperiodic spin chains with the ground state in an ASP, the two-point correlation functions of spins belonging to the same singlet and the mutual information of two blocks of consecutive spins are discussed in Sec. 4 and Sec. 5 respectively. With the aim of defining aperiodic spin chains with a large number of local DOFs, two generalizations of the models considered in [25] are worked out in Sec. 6. The first is obtained by arranging the local spin DOFs into multiplets of the dihedral group, which is the symmetry group of the truncated tiling inducing the aperiodicity of the chain. The second is achieved by considering aperiodic chains with globally $SO(N)$ invariant Hamiltonians. In Sec. 7 we provide conclusive remarks and point out interesting future research directions. Further discussions and computational details are reported in Appendices A-C.

## 2 Regular hyperbolic tilings

We begin by introducing the concept of regular hyperbolic tessellations, which are geometric discretizations of two-dimensional hyperbolic space. We briefly discuss infinite tilings before

restricting ourselves to finite truncations. We explain that the remaining symmetries of these finite tilings are given by the dihedral group $D_n$ of order $n = p, q$, whose finite-dimensional irreps are known. Moreover, we review the systematic construction of these hyperbolic tilings via inflation rules [25, 31, 36], and we recall the aperiodic structure that these induce at the asymptotic boundary of finite tilings.

Starting from $(2 + 1)$-dimensional AdS spacetime in global coordinates $\{\rho, t, \phi\} \in \{[0, 1), \mathbb{R}, [0, 2\pi)\}$, we restrict to a constant time slice, which induces the following hyperbolic metric on to the resulting Poincaré disk

$$ds^2 = (2L_{\text{AdS}})^2 \frac{d\rho^2 + \rho^2 d\phi^2}{(1 - \rho^2)^2}, \tag{1}$$

where $L_{\text{AdS}}$ denotes the AdS radius, and in the limit $\rho \to 1$, we have the conformal boundary. Subsequently, we discretize this constant time slice using regular polygonal tilings. These tilings cover the complete Poincaré disk with regular polygons with equal internal angles and equal edges. These tilings can be characterized in terms of two positive integer numbers $\{p, q\}$, where $p$ is the number of edges of each polygon, and $q$ is the number of polygons around each vertex. In order for the tessellation to tile a hyperbolic space, we require $(p-2)(q-2) > 4$. A particular way of constructing the tiling is to start with a tile centered at the origin and then by adding layers of tiles towards the boundary. We denote this configuration a *polygon-centered* tiling. Note that this procedure is in such a fashion that it preserves the symmetry of the central tile. Alternatively, this construction can start from a vertex at the origin, leading to what we denote accordingly as a *vertex-centered* tiling. These two possible configurations of a tiling are depicted in Fig. 1. If this construction is iterated infinitely many times, and thus tessellates the entire Poincaré disk, then the symmetry group of the resulting $\{p, q\}$ tiling is given by the so-called hyperbolic triangle group $\Delta(p, q, 2)$ [28–30]. If one restricts to orientation-preserving transformations, these groups belong to a class known as *Fuchsian groups* [53]. These have been recently studied in condensed matter theory in view of developing a band theory for tight-binding models on hyperbolic lattices [37–41]. However, as was discussed in [25], we require a UV cutoff for any practical computations. This amounts to truncating the infinite tessellation to a tiling of a finite extent, i.e. composed of a finite number of layers. From the construction explained above, the symmetry group of such truncated tilings is the dihedral group $D_n$ defined by the presentation

$$D_n = \langle r, s | r^n = s^2 = e, rs = sr^{-1} \rangle, \tag{2}$$

where $e$ denotes the identity of the group. From the presentation (2) it is clear that the generators $r$ and $s$ implement a rotation by an angle $\frac{2\pi}{n}$ and a reflection with respect to a fixed axis, respectively. For polygon-centered tilings, $n = p$, while for vertex-centered tilings $n = q$. Note that we include reflection transformations in order to obtain a richer representation theory, as will be discussed in Sec. 6.2. All irreps of $D_n$ are known and are discussed in detail in Appendix A.

In [31], a mechanism for constructing hyperbolic $\{p, q\}$ tilings was introduced based on so-called *inflation rules*. The key idea is to construct the tiling concentrically by adding layer after layer of tiles. For this purpose, a binary labeling scheme is introduced for the vertices at the boundary of each finite tiling along the way. This scheme assigns the letter $a$ to a vertex with two neighbors within the same layer, and the letter $b$ to a vertex with three neighbors. This labeling allows to describe each inflation step, i.e., the addition of a layer, in terms of replacement or inflation rules

$$\sigma_{\{p,q\}} = \begin{cases} a \mapsto a^{p-4}b(a^{p-3}b)^{q-3}, \\ b \mapsto a^{p-4}b(a^{p-3}b)^{q-4}, \\ c \mapsto (a^{p-3}b)^q, \end{cases} \tag{3}$$

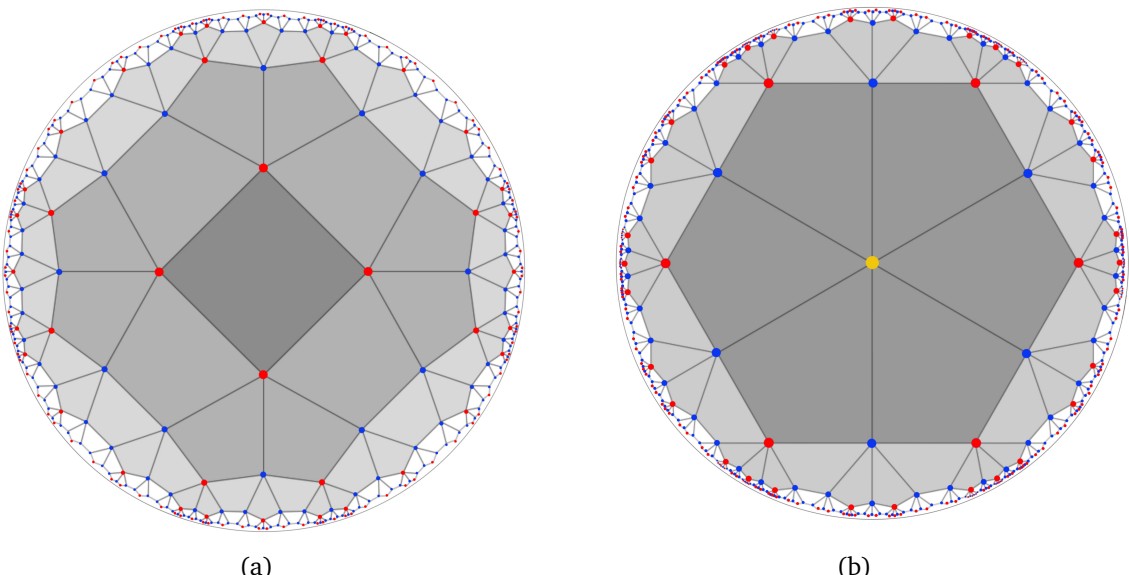

Figure 1: Construction of hyperbolic $\{p,q\}$ tilings through inflation rules. Vertices of type $a$, $b$ and $c$ are shown in red, blue and yellow, respectively. (1a): Polygon-centered $\{4,5\}$ tiling. Starting from a seed word $aaaa$ and applying the inflation rule $a \mapsto babab$ in (3), the vertices at the boundary of the second layer are obtained. (1b): Vertex-centered $\{4,6\}$ tiling. Starting from the center vertex of type $c$, the inflation rule $c \mapsto abababababab$ in (3) yields the vertices at the boundary of the first layer.

which dictate how each vertex at the boundary is substituted when a new layer is added to the tiling. Note that the vertex labeled by $c$ stands for *center* vertex, and thus appears only in the very first step of the inflation procedure of vertex-centered tilings. The application of the inflation rules (3) for the layered construction of tilings is shown in Fig. 1 in the exemplary cases of a polygon-centered $\{4,5\}$ and a vertex-centered $\{4,6\}$ tiling. In these figures, the vertex labels $a$, $b$ and $c$ are represented by red, blue and yellow dots, respectively, and each layer of tiles is colored in a different shade of gray. Technical subtleties may arise in this procedure for general values of $p$ and $q$ (e.g. the cases of $q = 3$ or $p = 3$) and we refer the reader to [25,31] for an exhaustive discussion. Based on the inflation rules (3), the inflation procedure starts from a seed word and applies iteratively the substitution rule. This iteration is performed a very large but finite number of times, such that the number of vertices on the boundary of the truncated tiling can be well-approximated by infinity. Up to a $\mathbb{Z}_n$ redundancy (cf. [25]), the inflation induces an asymptotic aperiodic sequence on the boundary, which we denote by $\mathcal{S}_{\{p,q\}}$. This aperiodic sequence at the boundary encodes information about the discretization in the bulk via the parameters $p$ and $q$. Some of the aperiodic sequences associated to $\{p,q\}$ tilings are known sequences, like the Fibonacci sequence for $\{5,5\}$ and the silver-mean sequence for $\{6,4\}$ [25,31,36].

As discussed thoroughly in [25], the inflation procedure can be practically implemented by the introduction of a so-called inflation matrix

$$M_{\{p,q\}} = \begin{pmatrix} (p-3)(q-3)+p-4 & (p-3)(q-4)+p-4 \\ q-2 & q-3 \end{pmatrix}, \tag{4}$$

for $p > 3, q > 3$, where the entries $(M_{\{p,q\}})_{ij}$ with $i,j \in \{a,b\}$ are the number of times the letter $i$ appears in the substitution word for the letter $j$ as per (3). Inflation matrices allow us to quantitatively characterize the resulting asymptotic aperiodic sequence $\mathcal{S}_{\{p,q\}}$. Specifically,

the largest eigenvalue $\lambda_+$ of $M_{\{p,q\}}$ describes the relative scaling factor of the asymptotic sequence after the application of a single inflation step. The components of the right and left eigenvectors, $\mathbf{v}_+ = (p_a, p_b)^t$ with $p_a + p_b = 1$ and $\mathbf{u}_+ = (l_a, l_b)^t$ such that $\mathbf{u}_+ \cdot \mathbf{v}_+ = 1$, describe the frequency and typical lengths of the letters $a, b$ in $\mathcal{S}_{\{p,q\}}$. For more details, we refer the reader to the discussion in [25, 31, 51, 54]. In particular, these parameters are crucial for the implementation of the SDRG on aperiodic spin chains, as was discussed in [25, 54], and as we will review in the following section.

## 3 Review of SDRG on aperiodic spin chains

In this section we review the main features of SDRG in aperiodic spin chains, highlighting how this technique is used to obtain results on the entanglement entropy in ASP [54, 55]. Within this task, we systematize the SDRG philosophy in a way that makes easier its application to models that have never been considered before in presence of aperiodic modulations. We focus on chains with a ground state given by an ASP at the SDRG fixed point. We point out that in these cases the dependence of the piece-wise entanglement entropy of a block of consecutive sites (and of its envelopes) on the sub-system size is the same for a general class of Hamiltonians. In particular, this dependence is provided solely by the details of the modulation [25, 54]. The explicit form of the aperiodic Hamiltonian enters only in a prefactor given by the entropy of a singlet $s_0$, which affects also the effective central charge.

As pointed out in [31, 36] and reviewed in Sec. 2, one can identify an aperiodic structure at the boundary of a finite $\{p, q\}$ tiling generated by the inflation procedure. This led the authors of [25] to argue that a theory defined at the boundary of these tilings should exhibit features of such aperiodicity. For this reason, aperiodic spin chains with modulations reflecting the bulk discretization have been proposed as boundary theories and their ground state and entanglement properties have been studied through SDRG. In the remainder of this manuscript, we focus on this class of theories, extending the analysis of [25]. In this section, we review the SDRG techniques [32, 33, 56, 57], which are the main computational tools exploited in the forthcoming sections.

We consider an infinite chain described by the following spin Hamiltonian with nearest-neighbors interaction

$$H_{\{p,q\}} = \sum_{i \in \mathbb{Z}} J_i \, h(\vec{\sigma}_i \cdot \vec{\sigma}_{i+1}; \theta_i), \tag{5}$$

where $\vec{\sigma}_i$ is a vector whose entries are spin degrees of freedom localized at site $i$ of the chain and $\cdot$ denotes the Euclidean scalar product between such vectors. The couplings $J_i \in \{J_a, J_b\}$ and $\theta_i \in \{\theta_a, \theta_b\}$ are spatially distributed following the aperiodic sequence $\mathcal{S}_{\{p,q\}}$. For the moment, we leave generic the nature of the spins, the function $h$ and its possible dependence on further parameters. Under general assumptions, the SDRG procedure can be applied to the Hamiltonian (5) independently of these features. Notice that the number of aperiodically distributed parameters in (5) can in principle be larger than two. Despite this, for clarity and convenience, here we consider only the couplings $J_i$ and $\theta_i$. Furthermore, we assume that the aperiodic Hamiltonian (5) is in a gapless regime [25]. For all the modulations given by the sequences $\mathcal{S}_{\{p,q\}}$, this is equivalent to requiring that the underlying homogeneous chain is critical [25, 51, 58].

The SDRG procedure is a real space renormalization group which is applied to non-homogeneous quantum chains. Our case of interest is when the spin chain is characterized by an aperiodic modulation, as the one of the Hamiltonian (5). Without losing generality, we can choose $J_b > J_a$ and focus on the coupling ratio $r \equiv J_a/J_b$. The main idea of the SDRG is that, at low energies, we can iteratively decimate out the blocks of consecutive spins linked

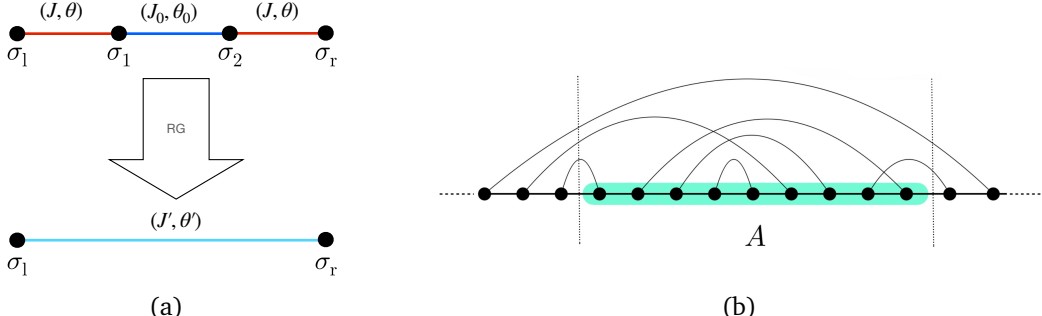

Figure 2: (2a) Fundamental decimation step in the SDRG of aperiodic spin chains. At low energies, the two spins $\sigma_1$, $\sigma_2$ coupled by a strong bond $(J_0, \theta_0)$ are decimated out of the chain and replaced by a renormalized coupling $(J', \theta')$ between the left and right spins $\sigma_1$ and $\sigma_r$. Whether $(J', \theta')$ is strong or weak in the renormalized chain must be determined a posteriori. (2b) Schematic depiction of the singlet distribution in an aperiodic singlet phase arising via the SDRG procedure. The entanglement entropy of a sub-region $A$ can be easily computed by counting the number of cut singlets shared between $A$ and its complement $B$. In the example of our figure, this number is equal to three.

by the stronger coupling $J_b$. This effectively renormalizes the coupling on the non-decimated bonds along the chain. This fundamental decimation step is shown in Fig. 2a for a block of two spins, albeit with a slightly different notation where $J_b = J_0$ and $J_a = J$, and similarly for the couplings $\theta_i$. Step after step, the decimation procedure induces a flow of the couplings, formally $r \to r' \to r'' \to \cdots \to r^*$ and $\theta_i \to \theta_i' \to \theta_i'' \to \cdots \to \theta_i^*$, with $i \in \{a, b\}$. The couplings $r^*$ and $\theta_i^*$ are the fixed points of the SDRG: if they depend on the initial values of $r$ and $\theta_i$, the modulation is said to be *marginal*, while, if they reach a certain value independently of the initial couplings, the modulation is called *relevant*. In the latter case, the SDRG is asymptotically exact, providing accurate predictions for the ground state of the chain [51,56].

Along SDRG procedures applied to generic modulations, the strongly coupled spin blocks can contain an arbitrary number of spins. In the following, we focus on the modulations whose SDRG allows only for blocks of two strongly coupled spins. The decimation procedure requires projecting the strongly-coupled block onto the ground state of the corresponding local Hamiltonian. In the case of the two-spins block, one has to solve the local Hamiltonian $h(\vec{\sigma}_1 \cdot \vec{\sigma}_2, \theta_b)$ introduced in (5). If the ground state of $h$ is non-degenerate, the decimated spin block is projected onto a singlet. In the case where only this type of decimation occurs, infinitely many iterations of the SDRG will lead to an aperiodicity-induced fixed point where the ground state of the aperiodic chain is given by the product of singlet states connecting spins at arbitrary distances. The system is then said to be in an *aperiodic singlet phase* [54]. A schematic depiction of such an ASP is given in Fig. 2b.

A remarkable feature of the SDRG when applied to aperiodic spin chains with Hamiltonian (5) is that the initial aperiodic sequence is recovered after a finite number of SDRG steps. Therefore, it stands to reason to implement the entire SDRG procedure in terms of the repetition of *sequence-preserving transformations*. If the original aperiodic sequence is recovered after $k$ SDRG steps, we say that the SDRG associated with that modulation is characterized by a *k-cycle*. In the forthcoming discussion, the SDRG flows characterized by 2-cycles play an important role. In these cases we can associate two $2 \times 2$ deflation matrices $M_1^{-1}$ and $M_2^{-1}$ to the two SDRG steps that combine together to provide the sequence-preserving transformation (see the discussion in Sec. 2). Because $M_1 \neq M_2$, their application allows to identify two different classes of singlets. More precisely, we can define *even* and *odd* singlets, according to

the number of SDRG steps after which they are decimated. The entire SDRG can be quantitatively described by the eigenvalues and eigenvectors of $M_1 M_2$ and $M_2 M_1$. A complete and detailed discussion is reported in [25]. Here, it is enough to recall the quantities that enter in the relevant expressions later in the manuscript. In our case of interest, $M_1 M_2$ and $M_2 M_1$ have the same eigenvalues; we call $\lambda_+^{(12)}$ the largest of them. We denote $(p_a^{(e)}, p_b^{(e)})^{\mathrm{t}}$ and $(p_a^{(o)}, p_b^{(o)})^{\mathrm{t}}$ the right eigenvectors associated to $\lambda_+^{(12)}$ of $M_1 M_2$ and $M_2 M_1$ respectively, normalized such that $p_a^{(i)} + p_b^{(i)} = 1$, with $i = e, o$. On the other hand, we call $(l_a^{(e)}, l_b^{(e)})^{\mathrm{t}}$ and $(l_a^{(o)}, l_b^{(o)})^{\mathrm{t}}$ the left eigenvectors associated to $\lambda_+^{(12)}$ of $M_1 M_2$ and $M_2 M_1$, respectively, with a normalization given by $l_a^{(i)} p_a^{(i)} + l_b^{(i)} p_b^{(i)} = 1$, with $i = e, o$.

In the following we list all the properties that an aperiodic spin chain with Hamiltonian (5) must satisfy such that the results found in [25] for the entanglement entropy in spin-1/2 aperiodic XXX chains can be generalized:

1. it must be critical, which for the modulations induced by the sequences $\mathcal{S}_{\{p,q\}}$ means that the homogeneous model underlying (5) must be gapless as well;

2. the modulation induced by $\mathcal{S}_{\{p,q\}}$ must be relevant;

3. the SDRG procedure on the aperiodic chain must be characterized by a two-cycle or a one-cycle modulation;

4. the ground state of (5) must be in an ASP.

Note that these requirements depend separately on the dynamics of $h$ in (5) and on the specific properties of the aperiodic modulation. Based only on the latter, it was shown in [25] that the following classes of $\{p, q\}$ modulations are the only ones for which 1-4 can be fulfilled: $\{6, q\}$ with $q \geqslant 4$, $\{5, 4\}$, $\{5, 5\}$, $\{3, 7\}$ and $\{3, 8\}$. For this reason, we focus on these modulations together with the general Hamiltonian (5) for the rest of our analysis.

In the cases where the specific Hamiltonian under consideration leads to the criteria 1-4 to be fulfilled, the entanglement entropy of a block of $L$ consecutive sites can be obtained as follows. In the ASP, the entanglement entropy is simply obtained by counting the number of singlets shared between the sub-system $A$ and its complement $B$ [59], cf. Fig. 2b. We call this number $n_{A:B}$ and it provides the total entanglement entropy in the singlet phase once it is multiplied by the entanglement entropy of an individual shared singlet, which is denoted by $s_0$. Notice that, since the aperiodic sequence of couplings is not homogeneous, $n_{A:B}$ depends on the position of the sub-system within the infinite chain. Therefore, we consider the number of shared singlets averaged over all the possible positions of $A$ and we call it $\bar{n}_{A:B}$. Strictly speaking, this quantity is a density, due to the infinite nature of the chain. For the arguments of this section, it is however simpler to refer to it as a total number. The resulting entanglement entropy is therefore an average entropy, but with a slight abuse of notation, we denote it simply as entanglement entropy, and it reads

$$S_A(L) = s_0 \, \bar{n}_{A:B}(L). \tag{6}$$

Due to the averaging over the aperiodic distribution of couplings, the quantity $\bar{n}_{A:B}(L)$ is a function of $p$ and $q$. In particular, it depends only on the aperiodic modulation of the chain and not on the explicit form of $h$ in (5). Its expression has been found in [25] and we report it in Appendix C. As discussed in detail in [25], the entropy in (6) has a piece-wise linear behavior as a function of the sub-system size. For two-cycles SDRG, two logarithmic envelopes can be identified for this function; they read [25]

$$S_{\mathrm{env}}^{(i)}(L) = \frac{c_{\mathrm{eff}}}{3} \ln L + \kappa_i, \qquad i = e, o, \qquad c_{\mathrm{eff}} = \frac{12 p_b^{(e)} l_b^{(e)} s_0}{\ln \lambda_+^{(12)}}, \tag{7}$$

where $c_{\text{eff}}$ is not a CFT central charge, but its definition is inspired by the results of entanglement entropy in critical systems [34, 60], see the discussion in [25]. Both the additive constant $\kappa_i$ and the effective central charge $c_{\text{eff}}$ depend on the details of the modulation, namely $p$ and $q$. In Appendix C we report the analytic expression of $\kappa_i$ and its derivation, making also clear why we label the two envelopes in (7) with the letters $e$ (even) and $o$ (odd). This provides an improvement of the results of [25], where such constants were simply fitted. Notice that in the case of a 1-cycle SDRG, the two envelopes in (7) coalesce into a single logarithmic curve [25, 54].

An insightful way to represent the ground state of an aperiodic spin chain is through a tensor network (TN). In [25], exploiting the SDRG procedure, a TN which reproduces exactly the ground state of a spin-1/2 aperiodic XXX chain was constructed. As a consequence, this TN encodes the symmetry and the entanglement structure of the ground state, allowing to represent it on the discrete hyperbolic geometry provided by the TN graph. We find it worth remarking here that this construction can be generalized, for instance, to the ground states of (5). Given an aperiodic spin chain with a relevant modulation determined by the sequence $\mathcal{S}_{\{p,q\}}$, the graph of the TN describing its ground state is exactly the same as the TN discussed in [25]. The reason is simply that the network structure is uniquely determined by the aperiodic sequence $\mathcal{S}_{\{p,q\}}$ through the SDRG. The only difference is the contribution of a single cut in the network to the entanglement entropy, or, in other words, the bond dimension, which depends on the explicit form of $h$ in (5).

In [25] the techniques reviewed in this section were applied to the spin-1/2 aperiodic XXX chain, whose Hamiltonian is given by (5) with $h(\vec{\sigma}_i \cdot \vec{\sigma}_{i+1}; \theta_i) = \vec{\sigma}_i \cdot \vec{\sigma}_{i+1}$, $i \in \mathbb{Z}$, and $\vec{\sigma}_i = (\sigma_i^{(x)}, \sigma_i^{(y)}, \sigma_i^{(z)})^{\text{t}}$ having the Pauli matrices as entries. For this specific model, the assumptions on the dynamics of the theory necessary for the validity of conditions 1-4 were argued to hold. In particular, the authors have shown that the the modulations induced by any sequence $\mathcal{S}_{\{p,q\}}$ are relevant for this model. In those cases leading to an ASP as ground state at the aperiodicity-induced fixed point, the entanglement entropy of a block of $L$ consecutive sites is given by (6) with $s_0 = \ln 2$ equal to the entanglement entropy of an EPR pair. In Sec. 6 we generalize this analysis to a broader class of Hamiltonians allowing for a large number of degrees of freedom on each site of the chain.

# 4 Two-point correlation functions in aperiodic singlet phases

In this section we present the computation of correlation functions of spin DOFs belonging to the same singlet in ASPs in general aperiodic Hamiltonians (5). We consider only two-point correlation functions in this work. We first briefly review the results obtained for the case of aperiodic modulations which induce a one-cycle SDRG, found in a condensed matter context in [51]. Then, we extend these results to the case of two-cycle SDRGs. For these modulations, we find that different two-point correlation functions are associated to each of the singlet generations discussed in the previous section. For each of these correlators we obtain a power-law decay with an exponent equal to unity. We observe that the only manifestation of $p$ and $q$ is encoded in the prefactor of the correlation function.

In [25], an expression for the length of discrete geodesics on hyperbolic tilings was obtained. We conclude the section by using these results in a direct discretization of known holographic relations between boundary correlation functions and bulk quantities in the continuum [61]. We find that the effective scaling dimension of boundary operators will increase after the discretization. Based on this, we provide a physical interpretation from the perspective of the bulk theory.

### 4.1 Review of correlation functions in one-cycle SDRG for ASP

We briefly review the analysis performed in [51] for the two-point correlation function in ASP that arise from one-cycle SDRGs. Though the analysis of [51] was performed for the spin-1/2 XXX Hamiltonian, it turns out to be more general and we can apply it to Hamiltonians of the form in (5) under the assumption that the criteria 1-4 are satisfied. Let us recall that in the cases where the SDRG leads to an ASP, the ground state of the chain is described by the product of spin singlets separated by arbitrary lengths. The dominant correlation between degrees of freedom in this ground state is the one between spins belonging to the same singlet, while all the other correlations are exponentially suppressed and thus comparably negligible. Specifically, the object we are interested in is

$$C^{\alpha\beta}(\Lambda) \equiv \overline{\langle \sigma_i^\alpha \sigma_{i+\Lambda}^\beta \rangle} = \delta_{\alpha\beta} C(\Lambda), \tag{8}$$

where $\Lambda$ denotes the typical length of a singlet and the last equality is due to the symmetry of (5). The function $C(\Lambda)$ was estimated in [51], and we briefly review it in the following.

Due to the nature of the ASP, the averaged correlation function $C(\Lambda)$ of two spin variables can be obtained by multiplying the correlation $c_0$ of two spins in a singlet by the density, or, more loosely, the number of singlets generated in a given SDRG step. Notice that $c_0$ can be straightforwardly computed from the local $h$ in (5). Let us denote by $\Lambda_j$ the typical length of a singlet in the $j$th generation. This will be inversely proportional to the number of spins $\rho_j$ that are part of a singlet in the same generation, roughly $\rho_j \sim 1/\Lambda_j$ [51]. The averaged correlation function of two spins in a singlet can be thus approximated as

$$C(\Lambda_j) \approx c_0(\rho_j - \rho_{j+1}) = \frac{c_0}{\Lambda_j}\left(1 - \frac{\Lambda_j}{\Lambda_{j+1}}\right). \tag{9}$$

For modulations which have a one-cycle SDRG, the quotient between typical lengths $\Lambda_j/\Lambda_{j+1}$, and thus, the entire bracket, tends to a constant $\sigma(p,q)$ as $j \to \infty$. Thus, the correlation function reads [51]

$$C(\Lambda_j) = c_0\sigma(p,q)\frac{1}{\Lambda_j}. \tag{10}$$

Notice how the dependence on the correlation length is the same as in the known result for continuum systems, having a scaling exponent equal to unity. The entire dependence of the result on the aperiodic modulation of the couplings is contained in the prefactor.

### 4.2 Correlation functions in two-cycle SDRG for ASP

As was pointed out in [25], there exists a large class of $\{p,q\}$ modulations which induce a two-cycle SDRG and lead to an ASP. This means that a sequence-preserving transformation $M_2M_1$ is composed by two distinct RG steps $M_1$ and $M_2$, and the SDRG procedure can be sub-divided into *even* and *odd* generations. Each of these generations has an associated typical length $\Lambda_j$ of the singlets present in it, as well as a typical concentration $\rho_j$ of singlets. A thorough derivation of these quantities can be found in [25], and we include a summary of the main expressions in Appendix B for the ease of the reader. Based, on these results, we now extend the results of [51] on the two-point correlation function to this two-cycle case.

Correlations are only non-vanishing between spins in the same generation. This means that in our treatment of the two-cycle case we have to consider two families of correlation functions, one for the even generations and one for the odd ones. This dependence may be encapsulated in the generic generation index $j \in \mathbb{N}$ whose value may be $j = 2k$ for the even generations and $j = 2k - 1$ for the odd ones. As in (9), we can approximate the correlation function by multiplying the contribution of an individual singlet $c_0$ by the relative number

(or density) of singlets in the $j$-th generation. Notice that this means we have to consider *exclusively* the singlets in the $j$-th generation, instead of the cumulative number of singlets decimated through SDRG in previous generations. Thus, we compute the density of singlets in the $j$-th generation by taking the difference between contiguous generations, obtaining

$$C(\Lambda_j) \approx c_0 \left( \rho_j - \rho_{j+1} \right) . \tag{11}$$

Note that the difference is with respect to the immediately next generation, and not with respect to the next generation of the same parity. This is because the later would overcount the number of singlets in the $j$-th generation by also including those that were present in the intermediate generation between $j$ and $j+2$. From (B.1)-(B.2) and (B.3)-(B.4), we can express the concentrations of singlets in terms of the typical singlet length, obtaining

$$\rho_{2k} = \frac{p_b^{(o)} l_b^{(o)}}{\Lambda_{2k}} , \quad \rho_{2k-1} = \frac{p_b^{(e)} l_b^{(e)}}{\Lambda_{2k-1}} . \tag{12}$$

Inserting this into the expression (11), and exploiting the fact that the products $p_b^{(e)} l_b^{(e)} = p_b^{(o)} l_b^{(o)} \equiv p_b l_b$ are independent of parity [25], we find

$$C(\Lambda_j) = \frac{c_0 l_b p_b}{\Lambda_j} \left( 1 - \frac{\Lambda_j}{\Lambda_{j+1}} \right) . \tag{13}$$

The quotient in the brackets tends to a constant which depends on the parity of $j$, namely

$$\frac{\Lambda_{2k}}{\Lambda_{2k+1}} = \frac{l_b^{(o)} \tilde{\lambda}}{l_b^{(e)} \lambda_+^{(12)}} \equiv \alpha^{(e)} , \quad \text{or} \quad \frac{\Lambda_{2k-1}}{\Lambda_{2k}} = \frac{l_b^{(e)}}{l_b^{(o)} \tilde{\lambda}} \equiv \alpha^{(o)} , \tag{14}$$

for $j = 2k$ and $j = 2k-1$, respectively. The auxiliary quantity $\tilde{\lambda}$ is explained in Appendix B and defined in (B.5). Notice in particular that these constants are dependent on the Schläfli parameters $p$ and $q$, effectively yielding $\alpha^{(e)} = \alpha_{(p,q)}^{(e)}$ and $\alpha^{(o)} = \alpha_{(p,q)}^{(o)}$. Our final result for the correlation function thus reads

$$C(\Lambda_j) = \begin{cases} \dfrac{c_0 l_b p_b \left( 1 - \alpha_{(p,q)}^{(e)} \right)}{\Lambda_{2k}} , & j = 2k , \\[3mm] \dfrac{c_0 l_b p_b \left( 1 - \alpha_{(p,q)}^{(o)} \right)}{\Lambda_{2k-1}} , & j = 2k-1 . \end{cases} \tag{15}$$

Let us highlight that the formula (15) also holds in the case where we have a one-cycle SDRG. In this case, $\tilde{\lambda}^2 = \lambda_+^{(12)}$, $l_b^{(o)} = l_b^{(e)}$ and therefore $\alpha^{(o)} = \alpha^{(e)}$, and thus (15) reduces to (10). Let us emphasize that the functional dependence of $C(\Lambda_j)$ on the distance between the spins we find in (15) is the same as the one found in [51]. Moreover, it also matches the continuum, homogeneous case, up to a logarithmic correction due to the enhanced symmetry of this fixed point [62]. We thus find that also in the case of a two-cycle SDRG the effect of the aperiodic modulation on the correlation functions manifests itself only in the constant prefactors.

As an explicit application of the results derive above, let us consider the case of $\{6, q\}$ modulations, which were shown in [25] to be an infinite family of two-cycle relevant modulations leading to an ASP. Closed formulas for $p_b^{(o)}, p_b^{(e)}, l_b^{(o)}, l_b^{(e)}, \tilde{\lambda}$ and $\lambda_+^{(12)}$ can be derived in this case for all $q \geq 4$ [25]. This allows us to investigate the behavior of the prefactors in (15) as a function of $q$. Their expressions are

$$1 - \alpha_{(6,q)}^{(e)} = \frac{2 \left( q^2 - \left( \sqrt{q^2 - 5q + 6} + 4 \right) q + 3 \sqrt{q^2 - 5q + 6} + 2 \right)}{3q - 10} , \tag{16}$$

$$1 - \alpha_{(6,q)}^{(o)} = \frac{2}{\sqrt{q^2 - 5q + 6} + q - 2} . \tag{17}$$

Two further explicit examples of correlation functions of two-cycle modulations leading to ASPs can be found in Appendix B.

Let us highlight that the ASP reached by our model through the SDRG describes a ground state with a factorized structure, where the density matrix factorizes into the tensor product of individual singlet density matrices entangling two spins. More precisely, all the $2n + 1$-point correlation functions are vanishing, while the $2n$-point correlation functions are non-zero only if all the spins in the correlators are pairwise coupled in the ground state density matrix. In the latter case, the contributions from the entangled spins factorize and one can employ (15) to compute the $2n$-point correlation function under consideration. Therefore, although the theory is interacting, the structure of the ASP leads to a factorization of higher-point correlations function into products of 2-point correlations functions. Moreover, since the local Hilbert space of the theory is a complex 2-dimensional space, the only relevant operators of the theory are the Pauli matrices and the identity. Thus, the characterization given above of the two-point functions between Pauli matrices fully determines all higher-point correlation functions of the theory.

### 4.3 Boundary correlation functions in holography

In the AdS/CFT correspondence, the two-point function of an operator $O_\Delta$ in the boundary CFT with scaling dimension $\Delta \gg 1$ can be written in terms of the length of the bulk geodesic $\gamma(x, y)$ connecting the two boundary points $x$ and $y$ as [61, 63]

$$\langle O_\Delta(x) O_\Delta(y) \rangle \propto e^{-\Delta \frac{\gamma(x,y)}{L_{\text{AdS}}}} . \tag{18}$$

Notice that here we are simplifying the formula since the geodesics should be evaluated in two cutoff dependent points which in the small cutoff limit go to $x$ and $y$. We skip these details here for simplicity.

Let us assume the validity of (18) also at the level of our discrete setup. In [25], an expression for the length $\gamma$ of a possible discrete geodesic $\Gamma$ connecting two boundary points was obtained as a function of the edge length $s(p, q)$, the asymptotic scaling factor $\lambda_+(p, q)$, cf. Sec. 2, and a parity function $v(p, q)$. We refer the reader to section 2 of the original source [25] for the precise definitions of these quantities and a detailed derivation of the discrete length. The important aspect is that we can replace the geodesic length $\gamma$ in (18) by its expression in the discrete case. In turn, this allows us to identify an *effective* scaling dimension which depends on the Schläfli parameters $p$ and $q$. Using the equations provided in [25], it is a straightforward to obtain the following

$$\langle O_\Delta(x) O_\Delta(y) \rangle_{\text{discr}} \propto \frac{1}{|x - y|^{2\Delta_{\text{eff}}(p,q)}} , \qquad \Delta_{\text{eff}}(p, q) = \frac{\Delta s(p,q) v(p,q)}{L_{\text{AdS}} \ln \lambda(p,q)} , \tag{19}$$

where $|x - y|$ denotes the number of sites between $x$ and $y$ on the discrete boundary.

In order to compare this effective scaling dimension with the original one, we consider the ratio $\Delta_{\text{eff}}/\Delta$ for different values of $p$ and $q$. Plugging in the explicit expressions for $s$, $v$ and $\lambda$ provided in [25], we find that this ratio is always greater than one, thus implying that the effect of the discretization is to increase the value of the scaling dimension. Let us provide a possible interpretation of this last result. In the continuum case, the bulk field which is holographically dual to $O_\Delta$ is localized around the continuum geodesic, but infinitesimal fluctuations of this trajectory will give correspondingly small corrections to the relation in (18). This is crucially different in the case of a discrete geodesic $\Gamma$, since any fluctuations of the trajectory would necessarily have to go along the $p - 1$ or $p - 2$ edges of a neighboring tile. Since the edge length of a hyperbolic tiling is of the order of the AdS radius, the smallest deviations $\delta\Gamma$ on the discrete geodesic will be of order $\mathcal{O}(L_{\text{AdS}})$. Thus, compared to the continuum case, allowed

fluctuations of the scalar field on the tiling need to be much larger in order to actually affect the discrete geodesic. In turn, this implies that the scalar field is more heavily localized on the discrete geodesic $\Gamma$ on the tiling than it would usually be on the continuum geodesic on $\mathbb{D}^2$. We can associate this localization to an effectively larger mass of the field. In the regime of large $\Delta$, we can approximate $\Delta \approx m L_{\text{AdS}}$, and thus a larger mass leads to a larger scaling dimension, which is exactly what (19) implies.

# 5 Mutual information in aperiodic spin chains

In this section, we compute the mutual information $I(A_1 : A_2)$ between two sub-systems $A_1$ and $A_2$ in ASPs. We first shortly review previous results put forward in [64], where the mutual information was investigated for random singlet phases. Then, we report new results on mutual information for ASPs, including a detailed analysis of one- and two-cycle modulations. We find that the mutual information of adjacent intervals exhibits a piece-wise linear behavior. Moreover, we obtain that the enveloping functions are a logarithm, matching the functional dependence expected from CFT calculations. The number of envelopes depends on the number of cycles of the SDRG for the corresponding modulation, as well as on the ratio of the sub-system sizes. The parameters of the modulation enter exclusively in the prefactor identified as the effective central charge, and in the non-universal constant (cf. (7)). We also consider the case of disjoint intervals separated by $d$ sites and compare it with holographic predictions from continuum AdS/CFT [9].

## 5.1 Mutual information in random spin chains

We briefly review the analysis of [64] on mutual information in random spin-$\frac{1}{2}$ XXX chains. The setup under consideration are two disjoint blocks $A_1$ and $A_2$ of the infinite chain, containing $L_1$ and $L_2$ consecutive sites, respectively. They are separated by a region $B_1$ containing $d$ sites, and the remaining complement region is denoted as $B_2$. The couplings $J_i$ are drawn from a random distribution instead of following an aperiodic sequence. The disorder can lead the system into a random singlet phase [65], similar to the ASP, but where the singlets are randomly distributed along the spin chain. The entanglement structure of this phase is the same in an ASP [54, 65], therefore the same formulae can be applied, once properly adapted, as explained in the next subsection.

The mutual information between $A_1$ and $A_2$ is defined as

$$I(A_1 : A_2) = S_{A_1}(L_1) + S_{A_2}(L_2) - S_{A_1 \cup A_2}(L_1, L_2, d).\tag{20}$$

The authors of [64] show that, in the singlet phase, the mutual information $I(A_1 : A_2)$ is proportional to the average number $\bar{n}_{A_1:A_2}$ of singlets shared between them. The proportionality constant is derived to be twice the entanglement entropy $s_0$ of an individual singlet. Concretely, the mutual information is found to be [64]

$$I(A_1 : A_2) = 2 s_0 \, \bar{n}_{A_1:A_2}.\tag{21}$$

Here, $\bar{n}_{A_1:A_2}$ is the average number of singlets shared between the two sub-systems in the ASP. The result in (21) is physically intuitive, since it states that all correlations between the sub-systems are encoded in the singlets connecting them. Computationally, however, (21) requires the evaluation of the entropy $S_{A_1 \cup A_2}$. Since the entanglement entropy of disjoint intervals is difficult to access in non-homogeneous spin chains, the authors in [64] re-wrote $S_{A_1 \cup A_2}$ in terms of a sum of entropies $S_\Omega(L_\Omega)$ of contiguous regions $\Omega = \{A_1 \cup B_1, B_1 \cup A_2, B_1, B_1 \cup A_1 \cup A_2\}$ with lengths $L_\Omega = \{L_1 + d, d + L_2, d, d + L_1 + L_2\}$. As explained in Sec. 3, the entanglement entropy

of a single interval is known in the case of aperiodic singlet phases, and is given by (6). Up to the proportionality constant $s_0$, this manipulation amounts to expressing the number $\bar{n}_{A_1:A:2}$ of singlets between $A_1$ and $A_2$ in terms of the numbers $\bar{n}_\Omega$ of singlets of contiguous regions with their corresponding complement. After a bit of algebra, the final result for the mutual information is [64]

$$I(A_1 : A_2) = S_{B_1 \cup A_1}(d + L_1) + S_{B_1 \cup A_2}(d + L_2) - S_{B_1}(d) - S_{B_1 \cup A_1 \cup A_2}(d + L_1 + L_2). \tag{22}$$

The main benefit of this form of the mutual information is that all the results obtained in [25] on the entanglement entropy of single intervals in ASPs can be directly implemented, as we explain in the next sub-section. Let us note that the entanglement negativity can be straightforwardly computed in ASPs as well by replacing $s_0$ with $n_0/2$ in (21), as done for chains with randomly distributed couplings in [64]. The parameter $n_0$ is the contribution of each singlet to the total negativity. A further analysis of this quantity is not within the scope of this work.

## 5.2 Mutual information in ASP

We now report in detail new results for the mutual information in ASPs obtained as fixed points of one- and two-cycle SDRG flows. Using (22) as our main quantity, we analyze the cases of adjacent and disjoint intervals separately.

**Adjacent intervals**

We begin with the mutual information for two adjacent intervals $A_1, A_2$, i.e. $d = 0$, as depicted in Fig. 3a. We can always write one sub-system length as a multiplicative factor of the other, e.g. $L_1 \equiv L$, $L_2 \equiv aL$, with $a > 0$. Then, (22) simplifies to

$$I(A_1 : A_2) = S_{A_1}(L) + S_{A_2}(aL) - S_{A_1 \cup A_2}((1 + a)L). \tag{23}$$

As was shown in [25], each of these individual terms exhibits a piece-wise linear behavior, with logarithmic enveloping functions given by (7). We can thus explicitly evaluate (23). The resulting mutual information, for the exemplary cases of $\{6, 4\}$ and $\{6, 6\}$ modulations, is given by the black lines in the panels of Fig. 3. The distinction between the panels will be explained shortly. As a general feature, we find that the mutual information $I(A_1 : A_2)$ again exhibits a piece-wise linear behavior. Moreover, we have full analytic control over the precise positions of the breaking points of this behavior. They arise from floor functions involved in the expressions (C.2) for the single-interval entanglement entropy, and we provide their derivation in Appendix C. One could expect that the mutual information exhibits an enveloping function given by the sum of the individual ones. However, a careful calculation reveals that this is a highly non-trivial statement. To understand this, note that the expression in (23) has in general three length scales, namely $L$, $aL$ and $(1+a)L$. Associated to each of these length scales there is a set of breaking points $x_k$ at which the corresponding term for the entropy $S_\alpha(x_k)$, with $\alpha = \{A_1, A_2, A_1 \cup A_2\}$, is exactly given by the logarithm function provided in (7). For adjacent intervals, we are indeed able to exploit the properties of the logarithm to manipulate terms like $\ln aL$ and $\ln(1 + a)L$ and extract an overall $\ln L$ factor. By keeping careful track of the non-universal additive constants $\kappa$ of the envelopes corresponding to all three length scales, we are able to find again a set of envelopes for the entire mutual information. Thus, similarly to (7), we find

$$I_{\text{env}}^{(i)}(A_1 : A_2) = \frac{c_{\text{eff}}}{3} \ln L + \beta^{(i)}, \quad i = 1, \ldots, k; \qquad k \in \{2, 3, 4, 6\}, \tag{24}$$

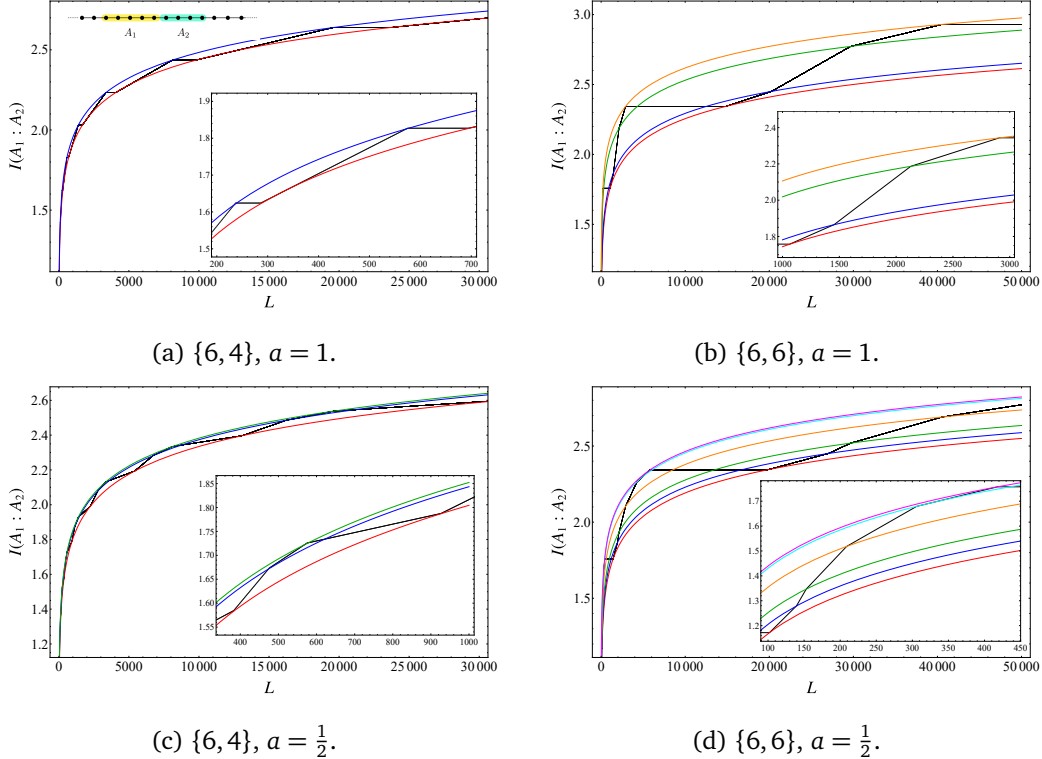

Figure 3: Mutual information $I(A_1 : A_2)$ for two adjacent intervals with $L_1 \equiv L$ and $L_2 \equiv aL$ in ASP obtained from different $\{p, q\}$ aperiodic modulations. For simplicity, we have chosen $s_0 = \ln 2$ in (21) for all plots. The four panels are ordered accordingly to the four cases of interest given in Table 1, exemplifying cases of one- and two-cycle modulations and equal and unequal sub-system sizes.

where we have defined a new set of non-universal additive constants $\beta^{(i)}$, whose number of elements is given by Table 1. We give the precise form of $\beta^{(i)}$ in Appendix C. Let us emphasize that the result in (24) exhibits the same functional dependence on the sub-system size as obtained from CFT computations [52, 66–68], but with a different multiplicative prefactor. It is this prefactor, together with the non-universal additive constant, where the effect of the modulation parameters $p$ and $q$ manifests itself. In fact, we find that the number of total envelopes is related to the number of length scales in the system, as well as to the cyclicity of the SDRG for the given modulation. This dependence is summarized in Table 1, and each case is correspondingly depicted in the individual panels of Fig. 3. As mentioned above, for unequal sub-system sizes $a \neq 1$ and one-cycle modulations, there exist three length scales, which coalesce to only two in the case of equal-length intervals with $a = 1$. Therefore, we observe three and two logarithmic envelopes in Fig. 3c and Fig. 3a, respectively, for the case of a $\{6, 4\}$ modulation. On the other hand, two-cycle modulations induce a doubling effect

Table 1: Number of enveloping functions, i.e different additive constants $\beta^{(i)}$, for the mutual information of two adjacent intervals with lengths $L_1 \equiv L$ and $L_2 \equiv aL$ in an ASP.

|            | one-cycle | two-cycle |
|------------|-----------|-----------|
| $a = 1$    | 2         | 4         |
| $a \neq 1$ | 3         | 6         |

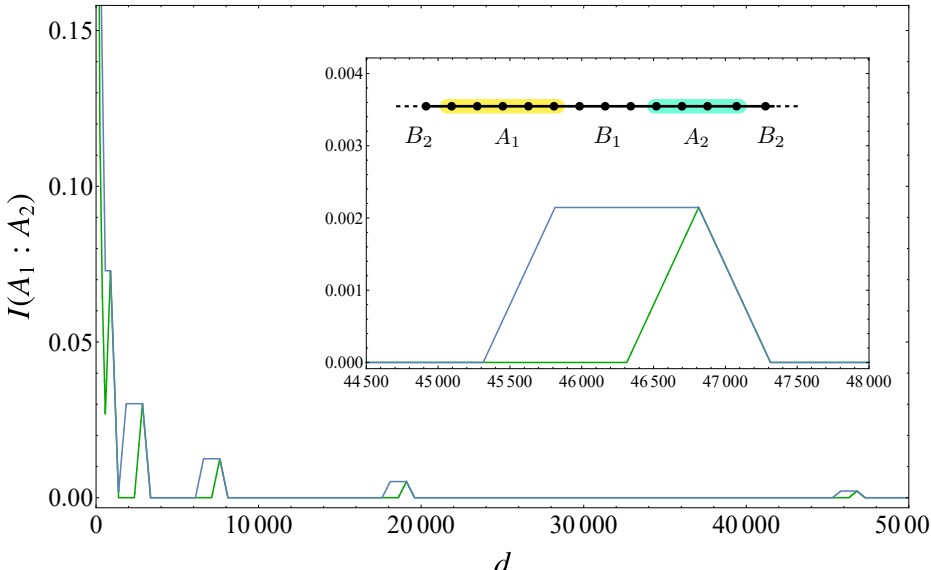

Figure 4: Mutual information $I(A_1 : A_2)$ for two disjoint intervals of equal (green line, $L_1 = L_2 = 500$) and unequal (blue line, $L_1 = 500$, $L_2 = 1500$, $a = 3$) length in the ASP obtained from a $\{6,4\}$ modulation. Inset: Peak of non-vanishing mutual information at around $d \approx 47000$ after a large regime of vanishing $I(A_1 : A_2)$. We have again chosen $s_0 = \ln 2$.

with respect to their one-cycle counterparts [25]. This is because now both even and odd generations each exhibit its own set of envelopes as in the one-cycle case. This is again precisely what we observe in Fig. 3b and Fig. 3d for the two-cycle $\{6,6\}$ modulation.

**Disjoint intervals**

We now turn to the case of two disjoint intervals, i.e. $d \neq 0$, as depicted in the inset of Fig. 4. The additional length scale $d$, which is necessary to compute entanglement entropies of the form $S(aL+d)$ in (22), complicates the analysis. Indeed, as explained in Appendix C, it enters additively in the floor functions in (C.2) and does not longer allow for the identification of enveloping functions. Nevertheless, since we have the analytical expression for each of the terms that constitute (22), we can evaluate the mutual information directly.

We are interested in the dependence of the mutual information on $d$. Thus, we fix a ratio of the sub-system sizes and analyze the mutual information as a function of the distance $d$. This is shown in Fig. 4 for the case of a $\{6,4\}$ modulation and both equal (green line) and unequal (blue line) sub-system sizes. The resulting behavior is a piece-wise linear decay of $I(A_1 : A_2)$. However, we find no enveloping functions. This, in turn, implies that a quantitative comparison with the behavior expected from CFT calculations [52,66–68] is not possible. In spite of this, we still have analytic control over the exact positions of the breaking points of the mutual information (see Appendix C). Notice that $I(A_1 : A_2)$ vanishes identically for certain ranges of $d$. Interestingly, the regions of vanishing mutual information are separated by "peaks" or "hills" of finite amplitude. The former are the case for equal sub-system sizes, while the latter appear for unequal intervals lengths. The surfacing of these regions becomes sparser as $d \to \infty$, and their amplitudes also decrease with increasing distance. Therefore, we find that the mutual information in ASPs indeed vanishes for large intervals of values of $d$, but it never remains zero for all $d$ greater than some critical value $d_c$. This is a unique feature of the ASP and originates from the existence of singlets at arbitrary scales. This means that singlet contributions to the mutual information can be expected at arbitrarily large distances $d$,

therefore preventing $I(A_1 : A_2)$ from remaining zero after any finite critical distance $d_c$. Note that this is in stark contrast with holographic results for mutual information. In particular, as a consequence of the celebrated Ryu-Takayanagi (RT) formula [6,7] for the holographic entanglement entropy, it is known that the holographic mutual information between two intervals in a CFT vanishes sharply after a critical distance [9]. This is because the corresponding RT surface in the bulk undergoes a first-order phase transition from a connected region to two disconnected geodesics.

# 6 Aperiodic spin chains with enhanced symmetries

In this section we apply ideas from continuous AdS/CFT to extend the studies of aperiodic spin chains. Exploiting symmetry arguments, we obtain two different classes of models. We discuss the SDRG properties and, for the case of the ground state of the chains being in an ASP, we discuss the features of entanglement entropy, mutual information and two-spins correlation functions.

## 6.1 Motivations from continuous AdS/CFT

In the paradigm of AdS/CFT, the operators defining the boundary theory transform covariantly under the conformal group, the latter being also the isometry group of the AdS spacetime. This property is an important manifestation of the holographic duality based on the symmetry matching between bulk and boundary. In order to achieve something similar in the discrete holographic setup introduced in [25] and reviewed in Sec. 2, we exploit the fact that the finite hyperbolic $\{p,q\}$ tiling has as symmetry group $D_n$ with either $n = p$ for polygon-centered tilings or $n = q$ for vertex-centered tilings. It is this property which motivates us to define and investigate an aperiodic spin chain with local spin DOFs arranged into multiplets transforming according to irreps of $D_n$. This is the goal of Sec. 6.2.

Another important aspect of standard continuous AdS/CFT is the large $N$ limit. Typically, the boundary theory is an $SU(N)$ gauge theory and the boundary fields belong to the adjoint representation of the $SU(N)$ gauge group. In order to incorporate a similar feature, in Sec. 6.3 we consider aperiodic spin chains with an $SO(N)$ global symmetry and spin DOFs defined as generators of $SO(N)$ in the fundamental representation. It is important to note, as will be explained in detail later, that in order to have a critical aperiodic spin chain, characterized by an ASP at the SDRG fixed point, we need to consider this specific symmetry group and representation rather the an $SU(N)$ symmetry with spins in the adjoint representation. The $SO(N)$ symmetry of the Hamiltonian allows to attain the limit $N \to \infty$. Though indeed similar to continuous AdS/CFT, let us stress that this limit is different, since the $SO(N)$ group for aperiodic spin chains plays the role of a global rather than a gauge symmetry.

## 6.2 $D_n$ symmetric spin chain Hamiltonians

In this subsection, we construct an aperiodic spin chain model incorporating the underlying symmetry group of the finite hyperbolic tiling, namely the dihedral group $D_n$ with $n = p, q$ as introduced in Sec. 2. We highlight that the dihedral symmetry in this setup denotes a symmetry which acts on the spatial coordinates of the DOFs. By exploiting the known representation theory of $D_n$, we construct an aperiodic Hamiltonian whose local DOFs are multiplets of irreps of $D_n$, generalizing the Hamiltonian of the spin-1/2 XXX chain. In the large $n$ limit, this construction allows us to access a regime where the number of local DOFs of the theory is large. We construct the Hamiltonian in such a way that the results about the correlation functions, entanglement entropy and mutual information discussed in the previous sections can be

readily adapted to this model.

Our starting point is the aperiodic spin-1/2 XXX chain described by the Hamiltonian

$$H^{\text{XXX}}_{\{p,q\}} = \sum_i J_i \, \vec{\sigma}_i \cdot \vec{\sigma}_{i+1} \,, \tag{25}$$

which is a specific instance of the Hamiltonian (5) where $h(\vec{\sigma}_i \cdot \vec{\sigma}_{i+1}, \theta_i) = \vec{\sigma}_i \cdot \vec{\sigma}_{i+1}$ and the three entries of the vector $\vec{\sigma}_i$ are the Pauli matrices. In order to construct a more general Hamiltonian whose spin DOFs are arranged into multiplets of the dihedral group, we first discuss how such multiplets transform under the action of the dihedral group $D_n$. Then, we combine these multiplets to obtain $D_n$-invariant terms, which are the building blocks of the Hamiltonian.

A practical feature of $D_n$ is that all of its irreps are known explicitly and they are either one- or two-dimensional. Moreover, there are only a finite number of irreps with each dimensionality. We provide a detailed discussion of them in Appendix A. Let us start with the one-dimensional irreps, of which there are four when $n$ is even, and only two when $n$ is odd. We denote the multiplets transforming under these one-dimensional irreps as one-dimensional multiplets $\vec{\sigma}_l$, with $l = 1, \ldots, 4$ or $l = 1, 2$, because each of them contains only one vector of Pauli matrices. Under the action of a generator $\lambda = \{r, s\}$ of the dihedral group (cf. (2)), these one-dimensional multiplets transform as follows

$$U_l(\lambda)\vec{\sigma}_{i,l}U_l(\lambda^{-1}) = \Phi_l(\lambda)\vec{\sigma}_{\lambda(i),l} \,, \tag{26}$$

where $U_l$ are unitary representations of the dihedral group acting on the local Hilbert space at the $i$-th site, and $\lambda(i)$ is the image of the $i$-th site through the applications of such generators. We provide the expressions for $\Phi_l(\lambda)$ in (A.1)-(A.4).

Although four one-dimensional irreps exist for $n$ even, we include only two of them in the following construction, namely those who transform trivially under the action of the generator $r$. Physically, this restriction implies that a rotation of the bulk, associated to a shift on the boundary, should not bestow an additional phase upon our DOFs. The irreps obeying this condition are $\Phi_1$ and $\Phi_3$ in (A.1)-(A.4) (see Appendix A). In particular, these are also the only two irreps present when $n$ is odd, further justifying our above constraint. We now adopt a unifying notation for these one-dimensional irreps, independently of the parity of $n$, by relabeling $\Phi_1$ and $\Phi_3$ as $\Phi_+$ and $\Phi_-$, respectively. Renaming accordingly the spin DOFs, and exploiting (26), we have the following action of the generators of $D_n$ on the one-dimensional multiplets $\vec{\sigma}_\pm$

$$U_\pm(r)\vec{\sigma}_{i,\pm}U_\pm(r^{-1}) = \vec{\sigma}_{r(i),\pm} \,, \qquad U_\pm(s)\vec{\sigma}_{i,\pm}U_\pm(s^{-1}) = \pm\vec{\sigma}_{s(i),\pm} \,. \tag{27}$$

If we interpret the generator $s$ as a parity transformation, we can regard $\vec{\sigma}_{i,+}$ as a scalar and $\vec{\sigma}_{i,-}$ as a pseudo-scalar, according to (A.1)-(A.4). Notice that $\vec{\sigma}_{i,+}$ and $\vec{\sigma}_{i,-}$ act non-trivially on two different Hilbert spaces and in this sense they are distinct degrees of freedom. Based on these one-dimensional multiplets, possible $D_n$-invariant terms are

$$\left(\vec{\sigma}_{i,l} \cdot \vec{\sigma}_{i+1,l}\right)^{\mathcal{P}_1} , \left(\vec{\sigma}_{i,l_1} \cdot \vec{\sigma}_{i+1,l_2}\right)^{2\mathcal{P}_2} , \qquad \mathcal{P}_1, \mathcal{P}_2 \in \mathbb{N}_0 \,, \text{ and } l_1 \neq l_2 \,. \tag{28}$$

With the aim of defining a generalization of the XXX spin chain with Hamiltonian (25), we will only consider terms with $\mathcal{P}_1 = 1$ and $\mathcal{P}_2 = 0$.

We now turn to the two-dimensional irreps of $D_n$, of which there exist $\lceil \frac{n}{2} \rceil - 1$, cf. Appendix A. We introduce the following variables at each site of the chain

$$\vec{T}_{i,m} \equiv \begin{pmatrix} \vec{\tau}_{i,m} \\ \vec{\tilde{\tau}}_{i,m} \end{pmatrix}, \qquad \vec{T}^\dagger_{i,m} = \begin{pmatrix} \vec{\tau}^\dagger_{i,m} & \vec{\tilde{\tau}}^\dagger_{i,m} \end{pmatrix} = \begin{pmatrix} \vec{\tau}_{i,m} & \vec{\tilde{\tau}}_{i,m} \end{pmatrix}, \tag{29}$$

and we define the product between $\vec{T}^{\dagger}_{i,m}$ and $\vec{T}_{i+1,m}$ as

$$\vec{T}^{\dagger}_{i,m} \cdot \vec{T}_{i+1,m} = \vec{\tau}_{i,m} \cdot \vec{\tau}_{i+1,m} + \vec{\tilde{\tau}}_{i,m} \cdot \vec{\tilde{\tau}}_{i+1,m}. \tag{30}$$

Analogously to the case of one-dimensional irreps, we may impose a transformation law on the DOFs in (29) under the action of the generators $r, s$ of the dihedral group,

$$U_m(r)\vec{T}_{i,m}U_m(r^{-1}) = \rho_m(r)\begin{pmatrix} \vec{\tau}_{r(i),m} \\ \vec{\tilde{\tau}}_{r^{-1}(i),m} \end{pmatrix} = \begin{pmatrix} e^{\frac{2\pi i m}{n}}\vec{\tau}_{r(i),m} \\ e^{\frac{-2\pi i m}{n}}\vec{\tilde{\tau}}_{r^{-1}(i),m} \end{pmatrix}, \tag{31}$$

$$U_m(s)\vec{T}_{i,m}U_m(s^{-1}) = \rho_m(s)\begin{pmatrix} \vec{\tau}_{s(i),m} \\ \vec{\tilde{\tau}}_{s(i),m} \end{pmatrix} = \begin{pmatrix} \vec{\tilde{\tau}}_{s(i),m} \\ \vec{\tau}_{s(i),m} \end{pmatrix}, \tag{32}$$

where $m = 1, \ldots, \lceil \frac{n}{2} \rceil - 1$ and $U_m$ are unitaries acting on the local Hilbert space associated to the $i$-th site. The matrices $\rho_m(r)$ and $\rho_m(s)$ are two-dimensional irreps of the dihedral group and are defined in (A.5) and (A.6). We denote the DOFs in (31) and (32) two-dimensional multiplets, in the sense that they contain two vectors of Pauli matrices. In the transformation rule (31), the component $\vec{\tilde{\tau}}_{i,m}$ is evaluated in $r^{-1}(i)$ after the application of the generator $r$, while $\vec{\tau}_{i,m}$ is evaluated in $r(i)$. Given that also the phases the two components pick up are one the inverse of the other, we can interpret $\vec{\tau}_{i,m}$ and $\vec{\tilde{\tau}}_{i,m}$ as a sort of analog of right and left movers with respect to the discrete rotations by $2\pi/n$ contained in $D_n$. For any fixed site $i$, the matrices $\vec{\tau}_{i,m}$ and $\vec{\tilde{\tau}}_{i,m}$ act non trivially on different Hilbert spaces and therefore we have $2(\lceil n/2 \rceil - 1)$ distinct two-dimensional Hilbert spaces.

In order to construct invariant terms involving two-dimensional multiplets, we now consider combinations of the type $\vec{T}^{\dagger}_{i,m} g^{(m)} \vec{T}_{i+1,m}$, where $g^{(m)}$ is a coupling matrix. Applying a transformation $U_m(\lambda)$, with $\lambda \in \{r, s\}$, on $\vec{T}^{\dagger}_{i,m} g^{(m)} \vec{T}_{i+1,m}$, we have

$$\vec{T}^{\dagger}_{i,m} g^{(m)} \vec{T}_{i+1,m} \rightarrow \vec{T}^{\dagger}_{i,m} \rho_m^{-1}(\lambda)g^{(m)}\rho_m(\lambda) \vec{T}_{i+1,m}. \tag{33}$$

The transformation in (33) leaves the term invariant if

$$\rho_m^{-1}(\lambda)g^{(m)}\rho_m(\lambda) = g^{(m)} \quad \Rightarrow \quad g^{(m)}\rho_m(\lambda) = \rho_m(\lambda)g^{(m)}, \tag{34}$$

where we have exploited that $\rho_m(\lambda)$ is unitary. The condition (34) means that in order for $g^{(m)}$ to be a coupling matrix associated to an invariant term, it has to commute with the generators of the group in the $m$-th two-dimensional irrep and therefore with all the elements of the group in that representation. By Schur's lemma, all such elements have to be proportional to the identity. Therefore, we can set without loss of generality $g^{(m)} = \mathbf{1}_m$ and therefore $\vec{T}^{\dagger}_{i,m} g^{(m)} \vec{T}_{i+1,m}$ becomes $\vec{T}^{\dagger}_{i,m} \vec{T}_{i+1,m}$. A similar argument can be given to show that terms of the kind $\vec{T}^{\dagger}_{i,m_1} \vec{T}_{i+1,m_2}$, with $m_1 \neq m_2$ are not invariant.

Having defined a set of DOFs transforming under irreps of $D_n$, we now provide an example of a $D_n$-invariant Hamiltonian. This Hamiltonian is defined over a Hilbert space with the following decomposition

$$\mathcal{H}_n = \bigotimes_{i \in \mathbb{Z}} \left[ \mathcal{H}_{i,+} \otimes \mathcal{H}_{i,-} \bigotimes_{m=1}^{\lceil n/2 \rceil - 1} \left( \mathcal{G}_{i,m} \otimes \tilde{\mathcal{G}}_{i,m} \right) \right], \tag{35}$$

where the DOFs $\vec{\sigma}_{i,\pm}$, $\vec{\tau}_{i,m}$ and $\vec{\tilde{\tau}}_{i,m}$ defined above act non-trivially only on $\mathcal{H}_{i,\pm}$, $\mathcal{G}_{i,m}$ and $\tilde{\mathcal{G}}_{i,m}$, respectively. All these Hilbert spaces are two-dimensional, and thus the local Hilbert space attached to each site of the chain, has dimension $2^{2\lceil \frac{n}{2} \rceil}$.

Now, considering only terms in (28) with $\mathcal{P}_1 = 1$ and $\mathcal{P}_2 = 0$ for the one-dimensional multiplets, and the terms (33) with $g^{(m)} = \mathbf{1}_m$ involving the two-dimensional ones, we propose the following Hamiltonian

$$H^{\text{dih}}_{\{p,q\}} = \sum_i \left[ J_i^{(+)} \vec{\sigma}_{i,+} \cdot \vec{\sigma}_{i+1,+} + J_i^{(-)} \vec{\sigma}_{i,-} \cdot \vec{\sigma}_{i+1,-} + \sum_{m=1}^{\lceil \frac{n}{2} \rceil - 1} J_i^{(m)} \vec{T}_{i,m}^{\dagger} \cdot \vec{T}_{i+1,m} \right], \qquad (36)$$

where all the couplings $J_i^{(+)}$, $J_i^{(-)}$ and $J_i^{(m)}$ are aperiodically modulated according to the sequence $\mathcal{S}_{\{p,q\}}$. By construction, the Hamiltonian (36) is invariant under the action of the dihedral group defined in (28) and (31)-(32). Moreover, by exploiting (30), it is easy to see that (36) is a sum of $2\lceil \frac{n}{2} \rceil$ XXX Hamiltonians of the kind (25), each of them acting non-trivially on only one of different Hilbert spaces appearing in the decomposition (35). The interesting feature is that, as explained in this section, the spin degrees of freedom can be arranged into multiplets of the dihedral group and the Hamiltonian (36) exhibits manifestly this structure, with the invariant terms obtained by constructing scalars out of the dihedral multiplets.

Let us remark that the Hamiltonian in (36) is not the most general dihedral-invariant Hamiltonian we can write down. Other invariant terms can be added; examples of these are obtained by simply taking powers of the terms in (36). Notice, moreover, that the Hamiltonian (36) can be easily generalized to include XXZ-like interactions by introducing an anisotropy along the $z$ direction of the spins. We do not further consider these generalizations since otherwise we would lose the physical insights and the analytical results we have under control. Nonetheless, we believe this to be an interesting path for future works.

The fact that the (36) can be written as a sum of independent aperiodic XXX Hamiltonians with the same modulation of the couplings allows us to readily apply the results for the entanglement entropy discussed in [54, 55] and [25], with the latter case specific to modulations induced by $\{p, q\}$ tilings. In particular, in presence of those relevant modulations that drive the ground state of spin-1/2 aperiodic XXX chains to an ASP, the entanglement entropy of a block of consecutive sites is given by (6) with $s_0 = \ln 2$ multiplied by the number of XXX-like contributions in (36), namely $2\lceil \frac{n}{2} \rceil$. As a consequence, the effective central charge associated to the model with Hamiltonian (36) is $2\lceil \frac{n}{2} \rceil c_{\text{eff}}$, with $c_{\text{eff}}$ given in (7). In other words, the central charge $c_{\text{eff}}$ gets rescaled by a factor proportional to the number of irreps of the dihedral group.

Given that the parameter $n$ labeling the dihedral group considered in this section can be either $p$ or $q$, varying it induces a modification on the aperiodic modulation of the couplings. Thus, in the perspective of the discrete holographic setting introduced in [25], tuning $n$ amounts to modify the underlying bulk geometries. In the limit $n \to \infty$, the boundary theory is characterized by a large amount of local degrees of freedom, while the bulk tiling becomes either a $\{\infty, q\}$ or a $\{p, \infty\}$. Let us stress that this limit has no counterpart in continuous AdS/CFT since, since the conformal algebra or, equivalently, the bulk symmetry group, have no parameter to be tuned. Nonetheless, given that we have analytic access to the infinite class $\{6, q\}$ with $q \geq 4$, we find it worth commenting on the results for this case. This is an important example given that the ground state of the corresponding aperiodic boundary spin chains is in an ASP and therefore the entanglement properties of these models are well understood. Naturally, we are only able to tune the parameter $q$ in this case and therefore we set $n = q$ for the following discussion. The effective central charge of spin-1/2 XXX chains has been computed in [25] and reads

$$c_{\text{eff}}(6, q) = \frac{6 - 2q + \sqrt{q^2 - 5q + 6}}{\ln\left(2q - 5 + 2\sqrt{q^2 - 5q + 6}\right)} \frac{6 \ln 2}{6 - 2q}. \qquad (37)$$

Notice that when $q \to \infty$, this $c_{\text{eff}}(6, q)$ decays to zero as $1/\ln q$. However, for the case of aperiodic spin chains with $D_q$ symmetry given by the Hamiltonian (36), the effective central

charge in (37) gets rescaled by a factor of $q$. This drastically changes its behavior for $q \to \infty$, which now exhibits a growth with $q$ instead of the decay mentioned above.

We conclude this section by explaining how the results of Sec. 4 on two-point correlation functions can be used to compute the spin-spin correlators of DOFs in the $D_n$-invariant Hamiltonian (36). Given that this Hamiltonian is a sum of $2\lceil \frac{n}{2} \rceil$ individual and independent XXX Hamiltonians with spin-1/2 $SU(2)$ DOFs, all of which are subject to the exact same $\{p, q\}$ aperiodic modulation, the behavior of the correlation functions of spins in the Hamiltonian (36) remains unchanged. This is because correlators are local objects that only relate two degrees of freedom which act on their corresponding Hilbert space. The only difference in the case of a $D_n$-invariant Hamiltonian like (36) is that we now have $(\lceil \frac{n}{2} \rceil + 1)$ different families of local DOFs, corresponding to the total number of irreps of $D_n$. Notice however, that only DOFs belonging to the same component of the same multiplet (either the one- or the two-dimensional one) can have non-vanishing correlations. Moreover, within that same component, correlators are non-vanishing only for DOFs of the same entry, effectively setting $\alpha = \beta$. Examples of such correlation functions are $\langle \sigma_{i,+} \sigma_{i+r,+} \rangle$ and $\langle \tilde{\tau}_{i,m} \tilde{\tau}_{i+r,m} \rangle$. Thus, the disorder averaged two-point function can be generically computed as in (10) and (15), for one-cycle and two-cycle modulations, respectively. The coefficient $c_0 = -1$ remains untouched, since it only depends on the two-point function of Pauli matrices.

## 6.3  $SO(N)$ aperiodic spin chains

In the following, we consider aperiodic spin chains with a global $SO(N)$ symmetry. We provide explicit Hamiltonians in a parameter regime that fulfills all the assumptions discussed in Sec. 3. In these models, we find that the entanglement entropy of a block of consecutive sites and its envelopes are given by (6) and (7), respectively, with $s_0 = \ln N$. The same holds for the mutual information, where the results are the ones obtained in Sec. 5, again with $s_0 = \ln N$. As a consequence, the effective central charges of these models diverge logarithmically when $N \to \infty$, a property expected from counting DOFs.

We consider the aperiodic spin chain with Hamiltonian given by (5), where $h$ is chosen to be [69]

$$h(\vec{\sigma}_i \cdot \vec{\sigma}_{i+1}, \theta) = \cos\theta \, \vec{\sigma}_i \cdot \vec{\sigma}_{i+1} + \sin\theta \left( \vec{\sigma}_i \cdot \vec{\sigma}_{i+1} + \frac{2(\vec{\sigma}_i \cdot \vec{\sigma}_{i+1})^2}{N-2} \right), \qquad N > 2. \tag{38}$$

The local spin DOFs contained in the vector $\vec{\sigma}_i$ are the $N(N-1)/2$ generators of the group $SO(N)$ in the fundamental representation (see [70] for a pedagogical introduction to the representation theory of Lie groups), i.e.

$$\vec{\sigma}_i = \left( \sigma_i^{(1)}, \sigma_i^{(2)}, \ldots, \sigma_i^{\left( \frac{N(N-1)}{2} \right)} \right)^{\text{t}}. \tag{39}$$

The resulting Hamiltonian has an $SO(N)$ global symmetry, realized through the invariance under the following action of an unitary representation of $SO(N)$ on the spin DOF $\vec{\sigma}_i$

$$\vec{\sigma}_i \to U_{\vec{k}}^\dagger \vec{\sigma}_i U_{\vec{k}}, \qquad U_{\vec{k}} \equiv e^{\mathrm{i} \sum_{j \in \mathbb{Z}} \vec{k} \cdot \vec{\sigma}_j}, \tag{40}$$

where $\vec{k}$ is a vector whose $N(N-1)/2$ entries are real parameters. The invariance of the Hamiltonian given by (5) and (38) can be verified by noticing that it commutes with all the components of $\sum_{j \in \mathbb{Z}} \vec{\sigma}_j$. We stress that the Hamiltonian given by (5) and (38) is the most general $SO(N)$-invariant Hamiltonian with nearest-neighbors interaction [69,71]. Indeed, for any pair of spins, one can show that $(\vec{\sigma}_i \cdot \vec{\sigma}_{i+1})^n$ with $n > 2$ can be expressed in terms of $\vec{\sigma}_i \cdot \vec{\sigma}_{i+1}$ and $(\vec{\sigma}_i \cdot \vec{\sigma}_{i+1})^2$.

We assume that the coupling $\theta$ in (38) is homogeneous along the chain and therefore the only parameter aperiodically distributed is the hopping $J_i$ in (5). In the corresponding homogeneous case, namely when $J_i = J$ on all the sites, the phase diagram of the model has been studied in [71]. It has been found that for $\pi/4 \leqslant \theta \leqslant \pi/2$ and $-\pi/2 \leqslant \theta \leqslant -\pi/4$ the chain is critical. Assuming $\theta$ to lie in this range, the chain remains critical also for aperiodic modulations given by the sequence $\mathcal{S}_{\{p,q\}}$, thus fulfilling condition 1 in Sec. 3.

In [69], the spin chains with Hamiltonian formally given by (5) and (38) but characterized by randomly distributed couplings have been studied and the SDRG procedure has been adapted in presence of $SO(N)$ DOFs. In the following, we generalize that approach to aperiodic $SO(N)$-invariant spin chains. We restrict ourselves to modulations induced by the sequences $\mathcal{S}_{\{p,q\}}$ with $\{p,q\} = \{6,q\}, \{5,4\}, \{5,5\}, \{3,7\}, \{3,8\}$. This choice implies that SDRGs are characterized either by a two-cycle or by a one-cycle and that only two-spin blocks are decimated throughout the whole procedure. In order for the ground states of such systems to be in an ASP, the ground state of $h$ in (38) must be a singlet. Given the $SO(N)$ symmetry, the eigenstates of $h$ are arranged into degenerate multiplets given by the $SO(N)$ irreps in which the tensor product of two fundamental representations decomposes. For $SO(N)$, this product decomposition gives rise to an $(N+2)(N-1)/2$-dimensional, an $N(N-1)/2$-dimensional and a one-dimensional irrep, where the latter contains the state we call *singlet*. Note that $N = 4$ and $N = 8$ are pathological cases from the point of view of the tensor product decomposition, cf. [70]. However, these still contain the singlet state we are interested in and we therefore include them in our analysis. From this structure, it is possible that the singlet is the ground state of the local Hamiltonian in some regime of parameters. This turns out to be the case when $-3\pi/4 < \theta < \arctan\left(\frac{N-2}{N+2}\right)$ [69]. Thus, we can conclude that, when $-\pi/2 \leqslant \theta \leqslant -\pi/4$, the $SO(N)$-invariant aperiodic spin chains we consider here satisfy also conditions 3 and 4 in Sec. 3.

Let us remark that, from the mathematical point of view, the tensor product decomposition of self-conjugate irreps is guaranteed to contain singlets [70]. The fundamental representation of $SO(N)$, as well as the one of $SU(2)$ are self-conjugate. However, this does not hold for $SU(N)$ with $N > 2$. In that case, the tensor product of two fundamental representations does not lead to singlets and therefore Hamiltonians like (5) with $\vec{\sigma}_i$ containing $SU(N)$ generators in the fundamental representation cannot lead to ASPs.

In order to understand whether the modulations induced by $\mathcal{S}_{\{p,q\}}$ for the aforementioned $\{p,q\}$ pairs are relevant, we must consider the explicit coupling flows along the SDRG. Using second order perturbation theory, one finds that the decimation of a two-spin block leads to a renormalization of the couplings given by [69]

$$J' \cos\theta' = f_1(N) \frac{J^2(\cos\theta)^2}{J_0(g(N) - \tan\theta_0)}, \qquad J' \sin\theta' = f_2(N) \frac{J^2(\sin\theta)^2}{J_0(1 - \tan\theta_0)}, \qquad (41)$$

where

$$f_1(N) \equiv \frac{4}{N(N+2)}, \qquad f_2(N) \equiv -\frac{4}{N^2}, \qquad g(N) \equiv \frac{N-2}{N+2}, \qquad (42)$$

and $\theta_0$ and $J_0$ are the couplings in the two-spins block before the decimation, $\theta$ and $J$ are the couplings connecting the block to the rest of the chain and $\theta'$ and $J'$ are the renormalized ones (see Fig. 2). Notice that the renormalization of the couplings of the two terms in (38) induces an renormalization of $J$ and $\theta$ separately.

The change of the couplings along the SDRG for an $SO(N)$-invariant aperiodic spin chain can be obtained from (41) generalizing the results of [25, 51, 56]. As an example, we consider the modulation induced by the sequences $\mathcal{S}_{\{6,q\}}$, with $q \geqslant 4$. This class encompasses both one-cycle ($q = 4$) and two-cycle ($q > 4$) SDRGs. The details of the SDRG for these aperiodicities are worked out in Sec. 3.3 of [25]. Here, we adapt that discussion to the $SO(N)$-invariant

Hamiltonian given by (5) and (38), finding the following renormalization of the strong and the weak couplings after a sequence-preserving transformation

$$J'_a \cos \theta'_a = J_a^{q-1} J_b^{2-q} \frac{f_1(N)^{2q-5} (\cos \theta_a)^{q-1}}{(\cos \theta_b \, (g(N) - \tan \theta_b))^{q-2} \, (g(N) + B(N, \theta_a, \theta_b))^{q-3}} \, , \tag{43}$$

$$J'_a \sin \theta'_a = J_a^{q-1} J_b^{2-q} (\tan \theta_a)^{q-3} \frac{f_2(N)^{2q-5} (\sin \theta_a)^{q-1}}{(\cos \theta_b \, (1 - \tan \theta_b))^{q-2} \, (1 + B(N, \theta_a, \theta_b))^{q-3}} \, , \tag{44}$$

and

$$J'_b \cos \theta'_b = J_a^{q-2} J_b^{3-q} \frac{f_1(N)^{2q-7} (\cos \theta_a)^{q-2}}{(\cos \theta_b \, (g(N) - \tan \theta_b))^{q-3} \, (g(N) + B(N, \theta_a, \theta_b))^{q-4}} \, , \tag{45}$$

$$J'_b \sin \theta'_b = J_a^{q-2} J_b^{3-q} (\tan \theta_a)^{q-4} \frac{f_2(N)^{2q-7} (\sin \theta_a)^{q-2}}{(\cos \theta_b \, (1 - \tan \theta_b))^{q-3} \, (1 + B(N, \theta_a, \theta_b))^{q-4}} \, , \tag{46}$$

where

$$B(N, \theta_a, \theta_b) \equiv (\tan \theta_a)^2 \left( \frac{N+2}{N} \right) \frac{(g(N) - \tan \theta_b)}{(1 - \tan \theta_b)} \, , \tag{47}$$

and $f_1$, $f_2$ and $g$ are defined in (42). Recall that each sequence-preserving transformation can contain many strongly coupled block decimations at the same time. This is the reason why the relations in (41) must be properly combined in order to get (43)-(46). We refer the interested reader to [25, 51, 56] for more details. Notice that an effective aperiodicity along the SDRG arises for the coupling $\theta$ even if in the original chain it is homogeneous. The formulae (43)-(46) must be applied in all the decimations along the entire original chain. Applying these relations iteratively during the whole SDRG procedure, one obtains the flows of the couplings. We are particularly interested in the flow of the coupling ratio $r = J_a / J_b$ and of the $\theta$ parameters, and both these flows can be read off from (43)-(46). The angle $\theta$ has two SDRG fixed points that can be obtained by studying the following expression [69]

$$\tan \theta'_i = \left( \frac{f_2(N)}{f_1(N)} \right)^{2(q-k_i)+1} \left[ \frac{(\tan \theta_a)^2 (g(N) - \tan \theta_b)}{1 - \tan \theta_b} \right]^{q-k_i+1} \left[ \frac{g(N) + B(N, \theta_a, \theta_b)}{1 + B(N, \theta_a, \theta_b)} \right]^{q-k_i} \, , \tag{48}$$

with $i = a, b$ and $k_a = 3$ and $k_b = 4$. The fixed points of (48) provides the fixed point of the coupling $\theta$. Within the range $-\pi/2 \leqslant \theta \leqslant -\pi/4$, one finds an unstable fixed point when $\theta_a = \theta_b = -\pi/4$ and a stable fixed point at $\theta_a = \theta_b = -\pi/2$. As for the flow of the coupling ratio, the renormalization $r \to r'$ reads

$$r' = r \, C(q, N, \theta_a, \theta_b), \tag{49}$$

where the function $C(q, N, \theta_a, \theta_b)$ can be obtained from (43)-(46). Its precise expression is not particularly illuminating and thus we do not report it here. The important feature to highlight is that $C(q, N, \theta_a, \theta_b) < 1$ for any values $-\pi/2 \leqslant \theta_a, \theta_b \leqslant -\pi/4$, $q \geq 4$ and $N > 2$. This means that $r' < r$ and therefore, iterating the SDRG procedure, the coupling ratio flows to zero. Thus, we conclude that the modulations induced by the sequences $\mathcal{S}_{\{6,q\}}$ on the Hamiltonian given by (5) and (38) are relevant and they drive the system to an aperiodicity-induced fixed point, thus fulfilling condition 2 as well. The same results can be obtained for the other aperiodicities mentioned earlier in this section, namely the ones associated to the pairs $\{5,4\}$, $\{5,5\}$, $\{3,7\}$, $\{3,8\}$.

We would like to stress that these disorder-induced fixed points are attractive with respect to the coupling ratio, which is the parameter tuning the aperiodicity. In contrast, the fixed points of the homogeneous counterparts of the models are repulsive with respect to this parameter, in the sense that a slight deviation from homogeneity $r \neq 1$ already drives the system into a new, disorder-induced fixed point. This is a feature of both the $SO(N)$-invariant model considered in this section, as well as the aperiodic $SU(2)$ XXX model reviewed

in Sec. 3. A detailed explanation regarding the nature of the fixed points, both homogeneous and aperiodicity-induced, for the case of the XXX model is given in Sec. 3 of Ref. [25].

We now summarize our analysis above and explain how we can exploit our results to make quantitative predictions: For any value of $J_a$ and $J_b$, the ratio $r$ always flows to zero (cf. (49)), while the coupling $-\pi/2 \leqslant \theta \leqslant -\pi/4$ stays at $\theta = -\pi/4$ if that is its initial value, otherwise it flows to $\theta = -\pi/2$. Therefore, for both these SDRG fixed points, the aperiodic chain with Hamiltonian given by (5) and (38) and with the modulations mentioned above satisfies all conditions 1-4 in Sec. 3. As a consequence, the entanglement entropy in this system is given by (6) and the mutual information by (21). The only parameter yet to be specified is the value of $s_0$. A straightforward computation yields $s_0 = \ln N$ [59]. This does not change the expression and the properties of entropy and mutual information as functions of $p$ and $q$ (see [25] and Sec. 5), but it indeed introduces an additional parameter, $N$, in the effective central charge given in (6). Interestingly, the effective central charge diverges logarithmically in the limit $N \to \infty$. This is different from the power law behavior in $N$ of the Brown-Henneaux central charge in standard continuous $AdS_3/CFT_2$ [4,47,72]. However, given the different interpretations of the parameter $N$ in the spin chain and in continuous AdS/CFT explained in Sec. 6.1, this apparent discrepancy is hardly a surprise.

Let us shortly discuss the two-point correlation functions of spins belonging to the same singlet in the ASP for the model considered in this section, following the discussion in Sec. 4. The general expression for the two-point function in ASPs is given by (15), where the only parameter depending on the explicit form of the Hamiltonian is $c_0$. For the case of $SO(N)$-invariant aperiodic spin chains, $c_0$ is obtained by computing the two-point function of the spins in the local Hamiltonian (38). It reads $c_0 = -2/N$. Thus, as discussed in Sec. 4, considering a different aperiodic Hamiltonian does not alter the dependence on $p$ and $q$ of the two-point functions with respect to the known case of aperiodic $SU(2)$ spin-1/2 XXX chains.

We conclude with a comment on the TN description of the ground states of the aperiodic $SO(N)$-invariant spin chains considered in this section. As emphasized at the end of Sec. 3, the aperiodic modulations considered here are relevant and they are uniquely determined by $\mathcal{S}_{\{p,q\}}$. Therefore, the structure of the TN graph which exactly reproduces the ground state is precisely the same as that discussed in detail in [25]. The only modification one needs to implement is adjusting the bond dimension of the tensors. Specifically, this bond dimension will now be given by $N$, and thus the entanglement associated to cutting a leg of the TN is accordingly modified. As shown in [25], whenever the tensors are associated to two-spin blocks decimated along the SDRG, they turn out to be perfect tensors and their entanglement is maximal. In the case of a TN reproducing the ASP of $SO(N)$-invariant aperiodic spin chains, this maximal entanglement is given by $\ln N$. Up to this modification, all other properties of the TNs derived in [25] remain unchanged. We refer to [73] for discussions on the general TN approach and to [25] for its application to aperiodic spin chains.

Furthermore, we would like to highlight that the TN thus represents an exact geometric description of the correlations present in the boundary ground state. This holds for all ASPs, not only the one of the $SO(N)$-invariant model considered in this section. In this view, the TN provides a geometrical interpretation for the behaviors found for the entanglement entropy in general and for the mutual information in particular. More precisely, the appearance of peaks in Fig. 4 is associated to the presence of TN legs running through the bulk and connecting spins in a singlet at arbitrarily large distances. Every time that such a leg is encountered, where each spin of the singlet lies in one of the intervals under consideration, it gives a contribution to the mutual information. Therefore, even though we find no sharp phase transition as is the case in continuum AdS/CFT, the behavior of a boundary quantity such as mutual information still possesses a geometric interpretation in the corresponding bulk TN.

## 7 Conclusions

We have analyzed infinite spin chains with couplings aperiodically modulated by the sequences characterizing the boundary of regular hyperbolic tilings of the Poincaré disk. These models were proposed in [25] as a first step towards a holographic duality involving a regular tiling of hyperbolic space in the bulk. The main results of this work are the following.

We have established a general framework for the analysis of aperiodic singlet phases (ASPs) at SDRG fixed-points via the conditions 1-4 given in Sec. 3. These include criticality, gaplessness, a particular cyclic behavior under SDRG transformations and the existence of an ASP. In particular, they imply that the entanglement entropy is directly given by (6). In addition, for an aperiodic Hamiltonian as in (5), where the explicit expression for the function $h$ determining the Hamiltonian and the nature of the spin DOFs are left generic, satisfying these conditions guarantees straightforward access to the correlation functions and to the mutual information. Using this framework, we obtain correlation functions of two spins in the same singlet in an ASP and generalize the findings of [51] to the cases when the SDRG on the aperiodic chain is characterized by two-cycles. We find that the two-point correlation function decays with a power law of the spin separation with exponent equal to one as given by (15). Moreover, a result from continuous AdS/CFT connecting the correlation function of boundary operators with scaling dimension $\Delta \gg 1$ to geodesics in the bulk [61, 63] is adapted to the discrete setup with the bulk given by a hyperbolic tiling. The resulting correlation function is modified with respect to the continuum case as reported in (19) and turns out to be associated with an effective scaling dimension larger than $\Delta$.

Furthermore, we obtain novel results, given by (22), on the mutual information of two blocks of consecutive sites in aperiodic spin chains. These are a generalization of the results of [64] to the presence of aperiodic modulations rather than random couplings. Similarly to what was found for the entanglement entropy [25, 54, 55], the mutual information exhibits a piece-wise linear behavior. For two adjacent sub-systems, the logarithmic enveloping functions (24) are found. These differ among them by an additive constant. These envelopes match the behavior computed in CFT [52, 66–68], with a prefactor reproducing the modulation-dependent effective central charge (6) already identified for the entanglement entropy in [25, 54, 55]. We determine that the number of envelopes depends on the ratio between the sub-system sizes and on whether the ASP is induced by a one-cycle or a two-cycle SDRG (see Table 1 on page 16). Moreover, we obtain explicit analytic expressions for the additive constants of such envelopes, cf. Appendix C. In the case of disjoint intervals, we find that the mutual information decays as a function of the number $d$ of sites separating the two sub-systems. More precisely, for increasing $d$, we observe increasingly long ranges where $I(A_1 : A_2)$ is zero, separated by non-vanishing peaks (see Fig. 4). Importantly, due to the presence in the ASP of singlets connecting spins at arbitrarily large distances, a value $d_c$ such that $I(A_1 : A_2) = 0$ for any $d > d_c$ does not exist and the phase transition known from continuum AdS/CFT is absent here.

Finally, motivated by two salient features of continuous AdS/CFT, namely matching of symmetries between bulk and boundary and the large $N$ limit, we study two models that satisfy the conditions 1-4 in Sec. 3. For the case that these conditions are satisfied, we obtain explicit results for the correlation functions, the entanglement entropy and the mutual information. The first of the two models is a spin-1/2 XXX chain with local spins arranged into multiplets transforming covariantly under the dihedral group $D_n$, cf. (36). We find that the effective central charge found in [25] is rescaled by a factor linear in $n$. For $\{6, q\}$ modulations in the large $n = q$ limit, the rescaled effective central charge is found to grow with $q$. The second model is the aperiodic Hamiltonian with a global $SO(N)$ symmetry given in (38). For this model we obtain an effective central charge which diverges as $\ln N$ for $N \to \infty$, reflecting the number of local DOFs.

Let us recall that, although we are primarily motivated by the program of discrete holography, the analysis and results obtained in the setup of this work should not be expected to reproduce holographic results. This can be argued from the fact the spin chains considered here do not allow for the notion of a strong-coupling limit, which is of crucial importance in holographic dualities. Also, the models under consideration do not have a gauge symmetry, as is usually the case in AdS/CFT. Instead, they have $SO(N)$ as a global symmetry group. Nevertheless, still guided by the motivation of finding discrete holographic dualities, we envision several possible interesting paths for further research, with the goal of introducing the above features:

First, a promising avenue consists of considering models such as chains [74] involving the Sachdev-Ye-Kitaev (SYK) model [48, 49]. At low energies and for large $N$, the $(0 + 1)$-dimensional SYK model of $N$ strongly-coupled, randomly interacting Majorana fermions is solvable, and can be described by an effective Schwarzian action. Moreover, in a particular regime the model is considered to be holographically dual [49, 75, 76] to Jackiw-Teitelboim (JT) dilaton gravity [77, 78]. In view of this, it seems promising to investigate the possibility of introducing aperiodicities into SYK-like chains. This would not only be novel from the perspective of spin chains, but it can also provide insights into potential discrete holographic duals by generalizing the known holographic features of the SYK model and of JT gravity. We look forward to exploring aperiodic SYK spin chain models as potential strongly-coupled boundary theories in the future.

Second, the TN description introduced in [25] together with its generalizations explained in this work, seems to have a yet unexplored potential for more quantitative analyses from the bulk point of view. In particular, it appears promising to consider aperiodic spins chains at finite temperature and their corresponding TN realization. In continuous AdS/CFT, thermal CFT states are holographically dual to black holes in asymptotically AdS spacetimes [5]. Therefore, a TN construction for aperiodic spin chains at finite temperature could describe the emergence of event horizons in the discrete bulk, without the necessity of ad hoc modifications of the network. We look forward to investigating this aspect of our work in the future.

Finally, the introduction of dynamical DOFs in the bulk is a pertinent next step in order to establish a proper discrete holographic duality. A particularly promising approach in this direction is provided by the setup of *edge length dynamics* [79], developed in the context of *p*-adic AdS/CFT [80–83]. Originally designed for tree graphs, this tool allows for fluctuating edge lengths in the graph describing a discretization of hyperbolic space. A discrete, graph-theoretic version of the Einstein-Hilbert action can be derived, which is used to assign probabilistic weights to different edge length configurations, in the sense of statistical mechanics. A generalization of this approach to hyperbolic tilings, which in particular include closed loops, offers an encouraging avenue for introducing bulk dynamics into the discrete holography program.

# Acknowledgments

We are grateful to Micha Berkooz, Latham Boyle, Elliott Gesteau, Justin Kulp, René Meyer and to Zhuo-Yu Xian for fruitful discussions. We particularly thank Zhuo-Yu Xian for bringing reference [69] to our attention. J.E. is also grateful to Ilya Gruzberg, Nele Callebaut and the participants of the workshop 'Random Geometry in Statistical Physics, Condensed Matter, and Quantum Gravity' at the Aspen Center for Physics for discussions.

**Funding information**   This work was supported by Germany's Excellence Strategy through the Würzburg-Dresden Cluster of Excellence on Complexity and Topology in Quantum Matter

- ct.qmat (EXC 2147, project-id 390858490), and by the Deutsche Forschungsgemeinschaft (DFG) through the Collaborative Research Center "ToCoTronics", Project-ID 258499086—SFB 1170. We further acknowledge the support by the Deutscher Akademischer Austauschdienst (DAAD, German Academic Exchange Service) through the funding programme, "Research Grants - Doctoral Programmes in Germany, 2021/22 (57552340)". This research was also supported in part by Perimeter Institute for Theoretical Physics. Research at Perimeter Institute is supported by the Government of Canada through the Department of Innovation, Science and Economic Development and by the Province of Ontario through the Ministry of Research, Innovation and Science. The work of J.E. was performed in part at the Aspen Center for Physics, which is supported by National Science Foundation grant PHY-1607611.

## A  Irreducible representations of the dihedral group

In order to enhance the symmetry the boundary theory of finite tiling by considering the copies of fields that transform according to the irreducible representation of the dihedral symmetry, we discuss the complete set of irreducible representations of this group. The dihedral group $D_n$ is defined in (2): here we list the complete set of its irreducible representations. For integer $n$, the number of irreducible representations (irreps) of $D_n$ is $\lceil n/2 \rceil + 3$ when $n$ is even and $\lceil n/2 \rceil + 1$ when n is odd. Here, $\lceil \cdot \rceil$ denotes the ceiling function. Among these, there are 4 one-dimensional irreps when $n$ is even and 2 one-dimensional irreps when $n$ is odd, while all the remaining $\lceil n/2 \rceil - 1$ irreps are two-dimensional [84, 85]. We only refer to all the inequivalent irreps, namely those irreps that are not relate to each other by a change of basis. The irreps are identified by their action on the generators. When $n$ is even, the one-dimensional irreps, denoted by $\Phi_l$, $l = 1, 2, 3, 4$, are given by

$$\Phi_1(r) = 1, \qquad\qquad \Phi_1(s) = 1, \qquad\qquad (A.1)$$

$$\Phi_2(r) = -1, \qquad\qquad \Phi_2(s) = 1, \qquad\qquad (A.2)$$

$$\Phi_3(r) = 1, \qquad\qquad \Phi_3(s) = -1, \qquad\qquad (A.3)$$

$$\Phi_4(r) = -1, \qquad\qquad \Phi_4(s) = -1. \qquad\qquad (A.4)$$

When $n$ is odd, only two out of the four in (A.1)-(A.4) are still irreps of $D_n$, namely $\Phi_1$ and $\Phi_3$.

We label the two-dimensional irreps by $m = 1, \ldots, \lceil n/2 \rceil - 1$. For $n$ either even or odd, their action on the generators are, in the basis where $\rho_m(r)$ is a diagonal matrix,

$$\rho_m(r) = \begin{pmatrix} e^{2\pi i m/n} & 0 \\ 0 & e^{-2\pi i m/n} \end{pmatrix}, \qquad\qquad (A.5)$$

$$\rho_m(s) = \begin{pmatrix} 0 & 1 \\ 1 & 0 \end{pmatrix}. \qquad\qquad (A.6)$$

Notice that all the irreps reported in (A.1)-(A.4) and (A.5)-(A.6) are unitary representations. These are used in Sec. 6.2 in order to define the transformation laws of the multiplets which are the building blocks of the $D_n$-invariant Hamiltonian (36).

## B  Details on two-cycle SDRG

In the two-cycle case, the original aperiodic sequence is only renormalized to itself after the combined application of two RG transformations $M_1^{-1}$, and $M_2^{-1}$. Keeping the notation of [25], the largest eigenvalue of $M_2 M_1$ is denoted as $\lambda_+^{(12)}$ and it is in general not the product of the

individual eigenvalues since the RG transformations do not commute. We denote as the 0-th generation the distribution of strong bonds on the original chain. The $j = 2k$-th ($k \in \mathbb{N}$) even generation of *singlets* is then obtained by applying $(M_1 M_2)^{-1}$ to the $k-1$-th even generation, while the $k$-th odd generation of singlets is obtained by applying $(M_2 M_1)^{-1}$ to the $k-1$-th odd one. This recursion relation necessitates the specification of a scaling factor $\tilde{\lambda}$ relating the singlet distributions of the first odd and even generations, i.e. between $k = 1$ and $k = 2$. This factor is not $\lambda_+^{(12)}$, since 1st and 2nd generation are related by a single application of $M_1$. It is also not the largest eigenvalue $\lambda_1$ of $M_1$, since $M_1$ does not, by itself, generate the asymptotic bond distribution of the even generations. It was shown in [25] how the factor $\tilde{\lambda}$ can be determined a posteriori. Notice, as mentioned above, that the typical length $\Lambda$ separating two spins in a singlet varies from generation to generation. In particular, we can assign a scaling of the typical length for the even and odd generations separately. This was derived in [25] to be as follows: Singlets in the $k$-th generation correspond to what were the strong bonds in the $k-1$-th generation and therefore their characteristic length $\Lambda_k$ reads

$$\Lambda_{2k-1} = l_b^{(e)} \left( \lambda_+^{(12)} \right)^{k-1} , \tag{B.1}$$

$$\Lambda_{2k} = l_b^{(o)} \left( \lambda_+^{(12)} \right)^{k-1} \tilde{\lambda} , \tag{B.2}$$

where $l_b^{(o)}$ and $l_b^{(e)}$ are the $b$-th components of the left eigenvectors associated to $\lambda_+^{(12)}$ of $M_2 M_1$ and $M_1 M_2$ respectively. Moreover, we have made use of the fact that the first generation of singlets corresponds to the distribution of strong bonds in the original chain, thus implying $\Lambda_1 = l_b^{(e)}$. This also explains the superscripts in (B.1) and (B.2), since the characteristic length of the *odd* generations is determined by the distribution of strong bonds one generation before, i.e. from an even one with $l_b^{(e)}$, and vice versa. In contrast, the concentration of singlets in the $k$-th generation reads

$$\rho_{2k-1} = p_b^{(e)} \left( \lambda_+^{(12)} \right)^{-k+1} , \tag{B.3}$$

$$\rho_{2k} = p_b^{(o)} \left( \lambda_+^{(12)} \right)^{-k+1} \tilde{\lambda}^{-1} , \tag{B.4}$$

where $p_b^{(o)}$ and $p_b^{(e)}$ are the $b$-th component of the right eigenvectors associated to $\lambda_+^{(12)}$ of $M_2 M_1$ and $M_1 M_2$ respectively. As shown in [25], the factor $\tilde{\lambda}$ may be derived via considerations of the concentration of singlets in the ASP and is found to be

$$\tilde{\lambda} = \frac{2 p_b^{(o)} \lambda_+^{(12)}}{\lambda_+^{(12)} (1 - 2 p_b^{(e)}) - 1} . \tag{B.5}$$

The above expressions play an important role in the computation of two-point correlation functions in ASPs, as was discussed in detail in Sec. 4 for the class of two-cycle modulations $\{6, q\}$. For the sake of completeness, we report here the results for the two-point correlation functions for two further modulations which fall in the regime of validity of (15). These modulations are associated to $\{5, 4\}$ and $\{3, 8\}$, both of which generate equivalent aperiodic sequences in the sense defined in [25]. Inserting the singlet typical lengths and concentrations given above for these two modulations into (15), the two-point functions read

$$C_{\{5,4\}}(\Lambda_j) = \begin{cases} \frac{c_0}{6} \left( 5 - \frac{7}{\sqrt{3}} \right) \frac{1}{\Lambda_{2k}} , & j = 2k , \\ c_0 \left( \frac{13}{2} - \frac{11}{\sqrt{3}} \right) \frac{1}{\Lambda_{2k-1}} , & j = 2k-1 , \end{cases} \tag{B.6}$$

and

$$C_{\{3,8\}}(\Lambda_j) = \begin{cases} c_0 \left( \frac{5}{2} - \frac{4}{\sqrt{3}} \right) \dfrac{1}{\Lambda_{2k}}, & j = 2k, \\[2mm] \frac{c_0}{6} \left( 9 - 5\sqrt{3} \right) \dfrac{1}{\Lambda_{2k-1}}, & j = 2k - 1, \end{cases} \tag{B.7}$$

respectively.

## C   Piece-wise behaviors and envelopes

In this appendix we report explicit computations for the logarithmic envelopes of the piece-wise entanglement entropy and mutual information in aperiodic spin chains in ASPs.

### C.1   Envelopes for the entanglement entropy

The sub-system is taken to be a block of $L$ consecutive sites in an infinite chain and the entropy is given in (6). In this case, it is enough to discuss only the average number of sites $\bar{n}_{A:B}(L)$, then recovering the entanglement entropy (and its envelopes) simply multiplying by $s_0$. The exact expression of the former as function of the sub-system size was first reported in [54] for one-cycle SDRGs and was later generalized to two-cycle SDRGs in [25]. In the following we focus on the latter case, since it includes the former as a particular case. For later convenience, we consider $\bar{n}_{A:B}(aL)$ with $a > 0$ real parameter. Its expression reads [25]

$$\bar{n}_{A:B}(aL) = 2 p_b^{(e)} l_b^{(e)} \nu(aL)$$

$$+ \frac{2aL \left( \lambda_+^{(12)} \right)^{-\nu(aL)/2}}{\lambda_+^{(12)} - 1} \times \begin{cases} \lambda_+^{(12)} \left( p_b^{(e)} + \tilde{\lambda}^{-1} p_b^{(o)} \right), & \nu(aL) \text{ even}, \\[2mm] \sqrt{\lambda_+^{(12)}} \left( p_b^{(e)} + \lambda_+^{(12)} \tilde{\lambda}^{-1} p_b^{(o)} \right), & \nu(aL) \text{ odd}, \end{cases} \tag{C.1}$$

where

$$\nu(aL) \equiv \left( \left\lfloor \frac{\ln \left( \frac{aL}{l_b^{(e)}} \right)}{\ln \lambda_+^{(12)}} \right\rfloor + \left\lfloor \frac{\ln \left( \frac{aL}{\tilde{\lambda} l_b^{(o)}} \right)}{\ln \lambda_+^{(12)}} \right\rfloor + 2 \right), \tag{C.2}$$

and $p_b^{(e)}, l_b^{(e)}, p_b^{(o)}, l_b^{(o)}$, and $\lambda_+^{(12)}$ are defined in Sec. 3 and their explicit expressions are given in Appendix B for some of the $\{p, q\}$ modulations leading to ASPs, together with the definition of $\tilde{\lambda}$. As discussed in [25], the quantity in (C.1) has a piece-wise linear behavior as a function of $L$. We call *breaking points* the values of $L$ at which non-analyticity occurs. Their expression is explicitly known and is given by $\Lambda_{2k-1}$ in (B.1) and $\Lambda_{2k}$ in (B.2) [25]. Whenever we want to refer to both the sequences of values, we denote them as $\Lambda_k$. These two series of breaking points give rise to two distinct logarithmic envelopes, which differ by an additive constant. The strategy to derive these envelopes, including the constant, is as follows: We first evaluate $\bar{n}_{A:B}(aL)$ for $L = \Lambda_{2k-1}$ and $L = \Lambda_{2k}$, which in particular requires to evaluate (C.2) at the same points. Subsequently, we check whether the $k$ dependence can be extracted from the floor functions in (C.2). If this happens and $\bar{n}_{A:B}(aL)$ in (C.1) is a smooth function either at $\Lambda_{2k-1}$ or at $\Lambda_{2k}$, then the corresponding expressions provide the envelopes.

Let us begin our analysis considering the *even* series of breaking points, namely $\Lambda_{2k}$. It is convenient to first evaluate $\nu(aL)$ for $L = \Lambda_{2k}$, obtaining

$$\nu(a\Lambda_{2k}) = 2k + \left\lfloor \frac{\ln a}{\ln \lambda_+^{(12)}} \right\rfloor + \left\lceil \frac{\ln\left(\frac{a l_b^{(e)}}{\tilde{\lambda} l_b^{(o)}}\right)}{\ln \lambda_+^{(12)}} \right\rceil \tag{C.3}$$

$$= 2\frac{\ln \Lambda_{2k}}{\ln \lambda_+^{(12)}} + 2 - 2\frac{\ln l_b^{(e)}}{\ln \lambda_+^{(12)}} + \left\lfloor \frac{\ln a}{\ln \lambda_+^{(12)}} \right\rfloor + \left\lceil \frac{\ln\left(\frac{a l_b^{(e)}}{\tilde{\lambda} l_b^{(o)}}\right)}{\ln \lambda_+^{(12)}} \right\rceil \tag{C.4}$$

$$\equiv 2\frac{\ln \Lambda_{2k}}{\ln \lambda_+^{(12)}} + 2 - 2\frac{\ln l_b^{(e)}}{\ln \lambda_+^{(12)}} + \xi_e, \tag{C.5}$$

where in the last step we have used the relation (B.2) between $k$ and $\Lambda_{2k}$. The crucial feature that is necessary for the existence of enveloping functions is that the dependence of $k$ can be extracted from the floor functions in (C.2). This is indeed what we find in (C.3)-(C.5). Notice that, once $a$ and the parameters of the modulation are fixed, $\nu(a\Lambda_{2k})$ has always a fixed parity. Thus, only one of the two branches in (C.1) occurs, determined by the value of $\xi_e$ defined in (C.5). Evaluating (C.1) at $L = \Lambda_{2k}$ and exploiting (C.5), we find

$$\bar{n}_{A:B}(a\Lambda_{2k}) = \frac{4 p_b^{(e)} l_b^{(e)}}{\ln \lambda_+^{(12)}} \ln \Lambda_{2k} + \tilde{\kappa}_e(a), \tag{C.6}$$

where

$$\tilde{\kappa}_e(a) \equiv 2 p_b^{(e)} l_b^{(e)} \left( 2 - 2\frac{\ln l_b^{(e)}}{\ln \lambda_+^{(12)}} + \xi_e \right) \tag{C.7}$$

$$+ \frac{2 a l_b^{(e)} \lambda^{-\xi_e/2 - 1}}{\lambda_+^{(12)} - 1} \times \begin{cases} \lambda_+^{(12)}\left(p_b^{(e)} + \tilde{\lambda}^{-1} p_b^{(o)}\right), & \xi_e \text{ even}, \\ \sqrt{\lambda_+^{(12)}}\left(p_b^{(e)} + \lambda_+^{(12)} \tilde{\lambda}^{-1} p_b^{(o)}\right), & \xi_e \text{ odd}, \end{cases}$$

and $\xi_e$ is defined in (C.5). The function (C.6) is one of the exact enveloping function of the piece-wise quantity $\bar{n}_{A:B}(aL)$ in (C.1). Multiplying (C.6) by $s_0$ defined in Sec. 3 and setting $a = 1$, we retrieve the envelope with $i = e$ in (7), where the additive constant can be read from (C.7) and is given by $\kappa_e \equiv s_0 \tilde{\kappa}_e(a = 1)$. A similar analysis can be performed to evaluate $\bar{n}_{A:B}(L)$ at $L = \Lambda_{2k-1}$, and we directly report the results, reading

$$\bar{n}_{A:B}(a\Lambda_{2k-1}) = \frac{4 p_b^{(e)} l_b^{(e)}}{\ln \lambda_+^{(12)}} \ln \Lambda_{2k-1} + \tilde{\kappa}_o(a), \tag{C.8}$$

where

$$\tilde{\kappa}_o(a) \equiv 2 p_b^{(e)} l_b^{(e)} \left( 2 - 2\frac{\ln\left(l_b^{(o)} \tilde{\lambda}\right)}{\ln \lambda_+^{(12)}} + \xi_o \right) \tag{C.9}$$

$$+ \frac{2 a l_b^{(o)} \tilde{\lambda} \lambda^{-\xi_o/2 - 1}}{\lambda_+^{(12)} - 1} \times \begin{cases} \lambda_+^{(12)}\left(p_b^{(e)} + \tilde{\lambda}^{-1} p_b^{(o)}\right), & \xi_o \text{ even}, \\ \sqrt{\lambda_+^{(12)}}\left(p_b^{(e)} + \lambda_+^{(12)} \tilde{\lambda}^{-1} p_b^{(o)}\right), & \xi_o \text{ odd}, \end{cases}$$

and

$$\xi_o \equiv \left\lfloor \frac{\ln a}{\ln \lambda_+^{(12)}} \right\rfloor + \left\lceil \frac{\ln\left(\frac{a\tilde{\lambda}\, l_b^{(o)}}{l_b^{(e)}}\right)}{\ln \lambda_+^{(12)}} \right\rceil . \tag{C.10}$$

This is the second envelop of the piece-wise function in (C.1). When $a = 1$ and (C.8) is multiplied by $s_0$, we obtain the envelop (7) of the entanglement entropy with $i = o$ and the additive constant given by $\kappa_o = s_0 \tilde{\kappa}_o(a = 1)$ (see (C.9)).

## C.2 Envelopes for the mutual information

In the following, we exploit the analytic expression of the envelopes (7) of the piece-wise entanglement entropy in order to derive the enveloping functions for the mutual information (22) of two blocks of consecutive sites. In the case of adjacent sub-systems made up of $L$ and $aL$ sites respectively, the formula (22) simplifies to (23). In the most general case of $a \neq 1$, we observe the presence of three length scales, namely $L$, $aL$ and $(a + 1)L$. For the two-cycle modulations we consider here, this implies the existence of six series of breaking points occurring at $L = \Lambda_{2k}$, $L = \Lambda_{2k-1}$, $L = \Lambda_{2k}/a$, $L = \Lambda_{2k-1}/a$, $L = \Lambda_{2k}/(a+1)$, $L = \Lambda_{2k-1}/(a+1)$ with $k \in \mathbb{N}$.

For each of these sets of breaking points, a similar computation to that above can be performed for evaluating the mutual information (23). In this way, we find six different enveloping functions, each of them associated to a different set of breaking points. Denoting the envelopes as $I_{\text{env}}^{(i)}(L, a)$, they read

$$I_{\text{env}}^{(i)}(L, a) = \frac{c_{\text{eff}}}{3} \ln L + \beta^{(i)}, \quad i = 1, \ldots, 6, \tag{C.11}$$

where the additive constants can be derived explicitly finding

$$\beta^{(1)} = \frac{s_0}{2} \left( \tilde{\kappa}_e(1) + \tilde{\kappa}_e(a) - \tilde{\kappa}_e(a+1) \right), \tag{C.12}$$

$$\beta^{(2)} = \frac{s_0}{2} \left( \tilde{\kappa}_o(1) + \tilde{\kappa}_o(a) - \tilde{\kappa}_o(a+1) \right), \tag{C.13}$$

$$\beta^{(3)} = \frac{s_0}{2} \left( \tilde{\kappa}_e(1/a) + \tilde{\kappa}_e(1) - \tilde{\kappa}_e(1+1/a) \right) + \frac{c_{\text{eff}}}{6} \ln a, \tag{C.14}$$

$$\beta^{(4)} = \frac{s_0}{2} \left( \tilde{\kappa}_o(1/a) + \tilde{\kappa}_o(1) - \tilde{\kappa}_o(1+1/a) \right) + \frac{c_{\text{eff}}}{6} \ln a, \tag{C.15}$$

$$\beta^{(5)} = \frac{s_0}{2} \left( \tilde{\kappa}_e(1/(a+1)) + \tilde{\kappa}_e(a/(a+1)) - \tilde{\kappa}_e(1) \right) + \frac{c_{\text{eff}}}{6} \ln(a+1), \tag{C.16}$$

$$\beta^{(6)} = \frac{s_0}{2} \left( \tilde{\kappa}_o(1/(a+1)) + \tilde{\kappa}_o(a/(a+1)) - \tilde{\kappa}_o(1) \right) + \frac{c_{\text{eff}}}{6} \ln(a+1), \tag{C.17}$$

with $\tilde{\kappa}_e$ and $\tilde{\kappa}_o$ defined in (C.7) and (C.9), respectively, $c_{\text{eff}}$ defined in (7) and $s_0$ introduced in Sec. 3 and dependent on the explicit form of the aperiodic Hamiltonian in (5). These results are reported in (24) with the slight change of notation $I_{\text{env}}^{(i)}(L, a) \equiv I_{\text{env}}^{(i)}(A_1 : A_2)$.

Let us remark that when $a = 1$, there are series of breaking points which coalesce and we are left with four series of breaking points, namely $L = \Lambda_{2k}$, $L = \Lambda_{2k-1}$, $L = \Lambda_{2k}/2$, $L = \Lambda_{2k-1}/2$, which in turn implies that the number of distinct envelopes reduces to four. This reflects the fact that only 2 length scales are present in the system. Moreover, when the SDRG is characterized by a one-cycle rather than a two-cycle (see Sec. 3), the series of breaking points $\Lambda_{2k-1}$ and the one $\Lambda_{2k}$ becomes a unique series and this halves the number of enveloping functions. This argument based on the analytic derivation of the envelopes justifies the contents of Table 1.

Let us conclude this Appendix with a discussion of the case of disjoint intervals $d \neq 0$. As reported in Sec. 5.2, no enveloping functions can be identified in this case for the mutual information. Besides this, we are particularly interested in the mutual information as a function of $d$ for fixed values of $L_1$ and $L_2$, cf. Fig. 4. Focusing on (22) and adapting the argument applied above for adjacent sub-systems, we find that the piece-wise mutual information has eight series of breaking points occurring at $d = \Lambda_{2k}$, $d = \Lambda_{2k-1}$, $d = \Lambda_{2k} - L_1$, $d = \Lambda_{2k-1} - L_1$, $d = \Lambda_{2k} - L_2$, $d = \Lambda_{2k-1} - L_2$, $d = \Lambda_{2k} - L_1 - L_2$, $d = \Lambda_{2k-1} - L_1 - L_2$ where $k \in \mathbb{N}$ and the expression of $\Lambda_k$ is given in (B.1) and (B.2). Notice, however, that plugging any of these series of breaking points into (22) implies the evaluation of entanglement entropies at arguments of the form $\Lambda_k$ + constant value. This becomes problematic when considering such arguments in the floor functions in (C.2). Indeed, due to the presence of logarithms, $\Lambda_k$ cannot be isolated and the dependence on $k$ cannot be extracted from the floor functions. As explained above, this is crucial for the derivation of the enveloping functions. Therefore, we conclude that no such envelopes can be identified for the mutual information in the case of disjoint intervals.

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
