# Peer review of "Aperiodic spin chains at the boundary of hyperbolic tilings"

_SciPost Physics, doi:SciPost Phys. 15, 218 (2023)_

## Round 1 · Referee Report · Anonymous (Referee 1) · 2023-7-28

Strengths

  • well written and clearly explained

  • lists points in favour of proposal and potential problems, this is valuable for the community

Weaknesses

  • somewhat incremental in nature

Report

This paper, together with ref [25] pursues the goal of establishing a version of discrete holography that rests upon an explicit microscopic boundary theory. The authors explore the possibility that such a boundary theory could be provided by so-called aperiodic spin chains (ASPs).

Apart from a well-written review of previous work on aperiodic tilings, the concrete results in this paper concern

  • a (so far) incomplete match of boundary two-point functions and properties of discrete bulk geodesics.

  • mutual information in ASPs: the authors define an effective central charge, by fitting their piece-wise constant mutual information has an envelope function fitting a logarithmic ansatz. Encouragingly the effective central charge so identified agrees with previous computations where available. On the other hand, a mutual-entropy phase transition expected from holographic arguments is found to not be present in the ASPs considered.

  • construction of new ASPs (with enhanced symmetries) motivated by expectations from continuous versions of AdS/CFT

I have no indications of any problems with the validity of the results reported.

On the other hand, the progress of the present paper appears to be more incremental in nature, and it is not entirely clear that it meets the high standards of SciPost regarding novelty and impact of published work. On the other hand, the authors do a good job to mention and collect evidence both in favour and in tension with the goal that ASPs could be boundary Hamiltonians of discrete holographic models. This is very valuable to the community and I would want to see these results published. However, the judgement of a second referee should be taken into account as well in making such this call.

  • validity: -
  • significance: -
  • originality: -
  • clarity: -
  • formatting: -
  • grammar: -

Author:  Pablo Basteiro  on 2023-08-23  [id 3917]

(in reply to Report 1 on 2023-07-28)

We would like to thank the reviewer for providing constructive and insightful feedback on the manuscript. The reviewer did not raise any specific points for revision, nor did they ask for any changes on the submitted manuscript. Our reply regarding the novelty of our results is contained in the general author's comments, since it was a pertinent point raised by both referees.

---

## Round 1 · Referee Report · Anonymous (Referee 2) · 2023-8-7

Strengths

1)The paper presented several analytical and exact results on these aperiodic spin chains , including a) 2 point correlation functions for 2 cycle SDRG b) some aspects of mutual information (particularly the logarithmic enveloping functions). These results hold independently of AdS/CFT.

2) The scaling of the mutual information with the degrees of freedom has a promising scaling.

3) The paper's review is quite clearly written.

Weaknesses

1) The connection between this tensor network and the usual form of AdS/CFT seems quite tenuous. a) The tensor network, while supposedly placed on the hyperbolic tiling, does not preserve much of the symmetries. b)The correlation functions also seem to bear very little relation with the actual geometry/connectivity of the tensor network.

c) The mutual information also does not bear much relation to the geometry. There is no phase transition in the mutual information observed as the separation of disjoint interval is changed.

The discrete geometry seems mostly responsible for generating the boundary sequence and therefore controls some symmetries of the final theory, but otherwise play little role in other aspects of the physics.

2) The correlation functions suggest that the conformal dimension of operators = unity. I do not understand why only one operator is explored and what this result implies about the spectrum of scaling operators in this spin chain in general-- does that mean it has other unexplored scaling operators or if this is the only one that is present in any 1 and 2-cycle aperiodic spin chain, so that the spectrum of scaling operator is trivial?

3) The framework of the aperiodic spin chain is established in reference [25]. The current paper is mostly focused on computing some results within the framework, applying various results already established in the previous paper. In that regard it would appear too similar to the previous paper and somewhat less impressive.

Report

I have a few questions, some of which are already discussed above.

1)As mentioned above, I have a question about correlation functions -- I have not understood if the spectrum of these spin chain is non-trivial.

2) The manuscript mentioned homogenous limit. Do these fixed points have homogenous limit so that they reduce to known critical models? I am not sure where the fixed points have parameters for one to tune to return to the homogenous model. Therefore I need some clarification whether they are related or when I should expect them to be related. In particular I doubt it is possible to recover the correct correlation functions of a homogenous critical spin models by counting singlets as in the paper. A clarification here would be very helpful.

3) There are more intuitive characterization of a theory than mutual information, such as the spectrum (which is part of the question above) and higher point correlation functions (say structure coefficients). Is there any reason why they are not discussed? Is it possible to include them in the paper?

On the other hand, these are novel models that exhibit interesting scaling and other entanglement behaviour that is probably of interest in their own right, with or without resemblance to the AdS/CFT. It is certainly progress that the paper obtains clean and exact results of these models. Therefore, I would still recommend the paper for publication, upon clarification of the above questions.

  • validity: high
  • significance: good
  • originality: good
  • clarity: high
  • formatting: good
  • grammar: excellent

Author:  Pablo Basteiro  on 2023-08-23  [id 3916]

(in reply to Report 2 on 2023-08-07)
Category:
answer to question

We thank the reviewer for their feedback and inquisitive questions on the manuscript. In the following, we address the raised points in the review.

1st point raised by the referee: Symmetries of the tensor network

Weakness #1 pointed out by the referee concerns the symmetries of the tensor network, and they remark that these do not fully match those of the tiling. Similar comments are made by the referee about other quantities like correlation functions and mutual information and their seemingly little relation to the tiling's discrete geometry.

Author's reply:

We agree with the referee and we are aware that a discrete holographic duality is not fully developed yet and the bulk interpretation of certain features of our proposed boundary theories remain to be determined. Still, we would like to highlight our intentions within this manuscript to address these issues. Our response to the referee's point is threefold:
First, we agree with the referee that we do not observe a standard phase transition in the mutual information as a function of the interval distance, as is usual in continuum AdS/CFT due to the geometry of the geodesics in the bulk. As the reviewer correctly pointed out, a holographic duality should be able to provide bulk-geometric interpretations for the behavior of boundary quantities like the mutual information. To partially overcome this, we would like to point out that we think of the tensor network as the actual geometric description of the correlations present in the boundary ground state. More precisely, we find that the tensor network we propose is indeed a precise geometrical representation of the aperiodic singlet phase ground state of the model. Moreover, our tensor network explicitly realizes the ground state of a known, explicit Hamiltonian. This is truly novel as compared to other approaches implementing TNs in AdS/CFT, where the boundary Hamiltonian is usually not known. We can therefore in fact provide a bulk interpretation to the behavior we find for the mutual information in terms of the TN graph structure. In particular, the behavior of the peaks appearing in Fig. 4 of the manuscript can be associated with the presence of individual TN legs running through the bulk and connecting intervals of the boundary at ever so large values of the distance, thus giving a contribution to the mutual information. We have included a paragraph emphasizing this interpretation at the end of Sec. 6.3 on page 27 of the revised manuscript.
Second, the tensor networks mentioned in the manuscript exhibit the same structure as those thoroughly investigated in Ref. [25]. Given that we consider large yet truncated hyperbolic tilings, their symmetry group is smaller than that of the infinite tiling. This is because the inflation procedure introduces a preferred direction, radially, which fixes the central tile or vertex as a reference point. Indeed, the symmetry group that survives the tessellation for finite tilings is the cyclic group $\mathbb{Z}_p$ or $\mathbb{Z}_q$ that leaves the central tile or vertex invariant. In this regard, the tensor networks indeed enjoy the same symmetries as the underlying truncated tiling, as was explained in more detail in Sec.~5.4 of Ref.~[25], to which we indeed refer in the manuscript.
Third, in order to involve the geometry of the bulk in a more direct way into the boundary theory, we have also followed an approach much in the spirit of AdS/CFT, where we attempt to imprint the symmetries of the bulk into the boundary. If one allows for orientation-reversion transformations on the truncated tiling, the remnant symmetry group is enhanced to the dihedral group $D_p$ or $D_q$, as is explained in Sec. 6.2 of the manuscript. It is precisely for this reason that we introduce $D_n$-invariant Hamiltonians, where $n=p,q$, which by construction enjoy the same symmetries as the bulk tiling.

2nd point raised by the referee: Operator spectrum and correlation functions

In the weaknesses' point 2, as well as in the report's points 1 & 3, the referee inquires about the spectrum of possible operators in the proposed theory, as well as the behavior of higher-point correlation functions of such operators. The reviewer states that these quantities typically provide a more intuitive characterization of the theory, and it is unclear to them whether these are trivial or not in the theory considered in the manuscript.

Author's reply:

We certainly agree with the reviewer in the sense that the behavior of correlation functions and the spectrum of the theory are quite constrained by the properties of the aperiodic singlet phase. The specific properties of this ground state allow access to analytic results at the cost of having a constrained factorized structure. We elaborate on this point and its consequences for the higher-point correlation functions and the spectrum below:

Correlation functions: The aperiodic singlet phase (ASP) reached by our model through the SDRG describes a ground state with very particular features which allow for analytical computations. Specifically, when the ground state of the aperiodic spin chain is in an ASP, the density matrix factorizes into the tensor product of individual singlet density matrices entangling spins at any spatial scale. Therefore, although the theory is interacting, the structure of the ASP leads to a factorization of higher-point correlation functions into products of 2-point correlation functions. More precisely, all the $2n+1$-point correlation functions are vanishing, while the $2n$-point correlation functions are non-zero only if all the spins in the correlators are pairwise coupled in the ground state density matrix. In the latter case, the contributions from the entangled spins factorize. A description of the 2-point correlation functions, given in Secs. 4.1 and 4.2 of the manuscript, is therefore enough to completely characterize the correlations in the ground state.

Operator spectrum: The local Hilbert space of the theory under consideration is a complex 2-dimensional Hilbert space. The Pauli matrices, together with the 2-dimensional identity operator, build a complete basis for this space, meaning that every operator can be decomposed into a linear combination of these matrices. There are no other possible local operators to be considered in this theory. Therefore, given the factorization of the higher-point correlation functions mentioned in the previous paragraph, we can fully determine all possible scaling dimensions in the ground state of the theory by computing the 2-point correlation function between different Pauli matrices, as performed in Secs. 4.1 and 4.2 of the manuscript.

Given the explanations above, it is clear that in the case of aperiodic spin chains with ASP as their ground state, the quantities suggested by the reviewer do not provide any more information as the ones presented in the manuscript. This is also the reason why we resort to quantities such as entanglement entropy and mutual information to provide a more detailed description of the ground states of these chains.
Nevertheless, we agree with the referee that these points might not be immediately clear from the discussion in the manuscript, and we have added explanatory sentences in this direction, much in the spirit of the above paragraphs, in Sec.~4.2 on pages 13-14 of the revised version of the manuscript. We have added a comment to clarify this important feature in the revised version of our manuscript.
Additionally, we would like to mention that a more thorough characterization of the spectrum as commented by the referee is strictly related to the fact of whether a continuum limit exists for aperiodic spin chains. We are not aware of any results in this direction at the moment but we cannot exclude it. This continuum limit would be very useful to further characterize, e.g., which are the scaling operators of the theory. We are aware of this interplay and certainly agree that an effective continuum description could be a very helpful future development in this regard.

3rd point raised by the referee: homogeneous limit

In point #2 of the reviewer's report, they ask for a clarification on the relation between the disordered chains considered in the manuscript and their homogeneous counterparts, which have a continuum limit description in general. We interpret these questions as asking for a clearer description of the differences between the fixed points of homogeneous and disordered spin chains.

Author's reply:

A detailed explanation regarding the properties of aperiodic disorder is given in Sec. 3 of Ref. [25], in particular the discussion around Fig. 6. In short, the strong-disorder fixed point reached via the SDRG which gives rise to the ASP as a ground state is an induced attractive fixed point which cannot be perturbed to retrieve back the homogeneous case. In contrast, the homogeneous fixed points of the theory are repulsive fixed point with respect to the disorder parameter. The aperiodic modulation is relevant in this sense, and one cannot tune a parameter to go back to the homogeneous case, since this would require the renormalization group to be a proper group instead of a semi-group.
Nevertheless, we agree with the referee that this can be pointed out more clearly in the manuscript. Therefore, we have included a referral to Ref. [25] in the revised version of our manuscript, together with a short explanatory sentence in this respect in Sec. 6.3 on page 25.

---

## Round 2 · Referee Report · Anonymous (Referee 2) · 2023-10-16

Report

Reading the changes the authors have made to the manuscript, I agree that several issues, including the factorisation of correlation functions and the spectrum of the theory, have been clarified.

As for the connection between the bulk tensor network geometry and the entanglement structure, I also agree with the authors clarification in the newly added paragraph, that there is some connection, except that when the bulk theory is not really in a semi-classical limit, it would be unlikely that geometrical objects such as geodesics would directly be related to either correlation functions or entanglement entropy. That they scale in the right way is perhaps as best as what one could hope for generally. In this light, I think that the paper has done what is possible in the current context and given the scarcity of exact results in describing quantum ground states using tensor networks, the paper generalising previous results to include more generic spins beyond spin 1/2 while providing many analytic results for these general cases would be of value to future studies of tensor networks in general.
I would therefore recommend the paper for publication in sci-post.

---

## Round 2 · Referee Report · Anonymous (Referee 1) · 2023-11-7

Report

The authors have sufficiently addressed the points raised in my previous report. I recommend publication in the present form.

---

## Round 2 · Author Response

Dear editor and dear referees,

we are grateful for the insightful comments based on a meticulous analysis of our manuscript. In order to address any queries about novelty as mentioned under `Weaknesses' in both reports, we would like to summarize and highlight the new results of this present paper in comparison to our previous paper Ref.[25]. These are as follows:

1) In the present paper we consider the large N limit of N -component spins as compared to spin 1/2 in our previous paper. For these we show that the effective central charge depends logarithmically on N , allowing for a regime where this effective central charge can be made large by taking N → ∞ . This is a further step towards usual holographic setups.

2) In the present paper we calculate the mutual information additionally to the entanglement entropy. We schematically show how to perform this analysis for generic aperiodic spin chains with aperiodic singlet phases as their ground states, and explicitly carry out computations in two example models. We find fully analytic results which exhibit an interesting behavior as a function of the distance which can be suitably explained in terms of both the properties of the aperiodic singlet phase ground state and the geometric properties of the corresponding tensor network.

Further replies to specific comments have been communicated to the referees individually.

We have revised our manuscript and replied to the referees' observations. A detailed account of the changes is given below, ordered as they appear in the manuscript. In the light of these clarifications and additions that address the referees' queries, we now consider the manuscript ready for publication in SciPost.

---

## Round 2 · List of Changes

List of changes:

  • On page 13-14, we have added an explanatory paragraph on the factorization properties of the aperiodic singlet phase and how these allow for the computation of higher-point correlation functions.
  • On page 26, we have included a paragraph elaborating on the properties of the homogeneous and disorder-induced fixed points with respect to the disorder parameter. We have also added a referral to Ref.~[25] for more details.
  • On page 27, we have added a paragraph on the interpretation of the bulk tensor network as a an explicit geometric description of the aperiodic singlet phase on the boundary, together with an explanation of the behavior found for the mutual information in terms of the tensor network geometry.

---

## Editorial Decision

published